# FRAMING THE GAME: HOW CONTEXT SHAPES LLM DECISION-MAKING

## ABSTRACT

Large Language Models (LLMs) are increasingly deployed across diverse contexts to support decision-making. While existing evaluations effectively probe latent model capabilities, they often overlook the impact of *context framing* on perceived rational decision-making. In this study, we introduce a novel evaluation framework that systematically varies evaluation instances across key features and procedurally generates vignettes to create highly varied scenarios. By analyzing decision-making patterns across different contexts with the same underlying game structure, we uncover significant and specific contextual influence on LLM decision-making. Our findings demonstrate this variability is largely predictable, yet acutely sensitive to framing effects. These results underscore the urgent need for dynamic context-aware evaluation methodologies to ensure reliable LLM deployment in real-world applications, and provides initial directions for their construction.

## 1 INTRODUCTION

Large Language Models (LLMs) have proved remarkably capable of generating human-like text, but their strategic decision-making in multi-agent settings remains an active area of investigation (Zhang et al., 2024). Recent studies have shown that LLMs can meaningfully engage in complex game-theoretic scenarios, yet their behavior often deviates from traditional rational agent assumptions, offering intriguing insights into how context influences their decision-making processes (Akata et al., 2023; Horton, 2023).

This paper examines how narrative contextual framing affects an LLM's strategic and rational decision-making. We introduce a novel evaluation framework that generates decision scenarios across diverse user-defined contexts while maintaining consistent underlying game logic. By preserving the same game structure across scenarios, we can isolate and analyse the potential contextual influence on LLM decisions. Additionally, our approach of generating multiple scenarios helps us control for the inherent stochasticity of LLM responses.

We then apply this framework across over 20 state of the art LLMs. including both open- and closed-source models at a variety of sizes. We specifically explore the single-shot Prisoner's Dilemma (PD) as our canonical game due to its foundational role in studying strategic decision-making and cooperation. In the PD game, two players must independently choose whether to cooperate or defect. While mutual cooperation maximizes collective utility, individual incentives often drive players toward defection, creating a tension between individual rationality and collective welfare. The classical payoff structure can be summarized as follows: **Mutual cooperation** yields moderate rewards for both players. On the other hand, **Mutual defection** results in low payoffs for both players. Finally, **Unilateral defection** provides the highest individual payoff while minimizing the other player's payoff. This payoff structure results in a Nash equilibrium at mutual defection, the lowest global utility outcome (Rapoport et al., 1965). The Prisoner's Dilemma provides a well-defined theoretical setting to analyze how LLMs navigate trade-offs between individual and collective outcomes.

Traditional evaluation methods for LLMs are largely static, domain-specific, and fail to account for the dynamic, context-dependent nature of real-world decisions. Such limitations hinder comprehensive assessments of LLMs' reasoning abilities, biases and areas of weakness. We adress this with the following primary contributions:

- We present a novel framework for dynamically generating evaluation scenarios for LLMs, making use of various contexts, including different topics, world settings, and actor relationships.

- Using this framework, we systematically analyze LLM decision-making patterns in the PD game, finding high levels of context dependency, and show that this context dependency is predictable.

- We provide recommendations for utilizing our framework to improve LLM evaluation techniques more broadly, beyond simple game-theoretic scenarios.

## 2 RELATED WORK

### 2.1 LLM EVALUATION

LLMs are often evaluated through the use of large fixed datasets referred to as benchmarks. These benchmarks range from general purpose question answering (e.g. MMLU (Hendrycks et al., 2020), AGIEval (Zhong et al., 2024)) to specialized benchmarks testing specific capabilities such as mathematics (Cobbe et al., 2021; Glazer et al., 2024), coding (Chen et al., 2021; Jimenez et al., 2024), or scientific knowledge (Lu et al., 2022; Rein et al., 2024). One issue with most such benchmarks is data contamination (Xu et al., 2024; Deng et al., 2024), which creates a potential mismatch between inflated benchmark results and real-world performance. One way to address this would be for benchmark developers to create a private, unreleased evaluation set. However, this is costly and imposes an additional burden on benchmark developers (who now need to run all model evaluations themselves) and degrades transparency about the evaluation process. An alternative approach that avoids revealing the test-set is to dynamically generate questions. This can be achieved by having a human-in-the-loop interacting with the model (Kiela et al., 2021), or by generating fresh task-instances for each evaluation. Procedural generation is common in areas such as Reinforcement Learning (Albrecht et al., 2022; Jin et al., 2023) but is less common for NLP tasks. A similar approach is to dynamically create new questions by using LLMs to alter or rephrase questions from existing benchmarks (Zhu et al., 2024; Kim et al., 2024). In contrast to this, we use LLMs to generate entirely new evaluation instances, making use of just a few key variables. This enables our methodology to easily produce much more diverse evaluation instances. Moreover, by maintaining the same underlying game structure, we can evaluate and compare responses programmatically without human input, making our method fast, cheap and scalable.

### 2.2 CONTEXT FRAMING, GAME THEORY, AND LLMS

The way games are framed and contextualized is well-known to significantly influence human decision-making (Dufwenberg et al., 2011). This effect is particularly pronounced in the Prisoner's Dilemma, where it has been well studied that contextual framing shapes players' choices in cooperative versus competitive scenarios (Liberman et al., 2004; Goerg et al., 2020; Columbus et al., 2020). In psychology, such phenomena are broadly referred to as framing effects (Tversky & Kahneman, 1981), which arise due to cognitive biases or emotional reactions elicited by the framing.

More recently, LLMs have been studied as participants in game-theoretic scenarios Duan et al. (2024). Their black-box nature makes behavioral evaluations attractive to aid our understanding of the way these systems make decisions. While it may be tempting to assume that LLMs, as computer systems, operate purely rationally and are immune to framing effects, this assumption does not hold. Trained on vast amounts of human-generated data, LLMs exhibit cognitive biases and frequently deviate from the rational agent model (Echterhoff et al., 2024). *Mozikov et al.* demonstrated how emotional prompting affects LLMs' decision-making in four classical game theory scenarios (Mozikov et al., 2024). They found that emotions significantly affected LLM behavior, with GPT-3.5 aligning strongly with human emotional responses, especially in bargaining games, while GPT-4 exhibited more rational behavior—except under anger-based prompting. Similarly, *Lore and Heydari* examined how game structure and contextual framing influenced decision-making in GPT-3.5, GPT-4, and LLaMA-2 (Lorè & Heydari, 2024). They observed that GPT-3.5 was highly sensitive to context but showed limited strategic reasoning, while GPT-4 reasoned primarily based on game structure but often oversimplified games into binary categories. LLaMA-2 exhibited a more nuanced understanding of game mechanics but remained sensitive to context framing While these studies provide valuable

insights, they are limited in scope and methodology. For instance, *Mozikov et al.,* varied only the emotional tag appended to prompts, keeping the game context constant as a simple two-player game with payoffs explicitly defined in the text (Mozikov et al., 2024). Similarly, *Lore and Heydari* explored only five static contexts and provide the payoff matrix directly, signaling to the model that the task given is explicitly a game Lorè & Heydari (2024). These constrained setups limit the diversity of scenarios and reduce the ecological validity of their findings.

In our work, we propose generating diverse vignettes drawn from contexts the model could see after deployment in the real world. By breaking context into multiple categories like topic and actor type we can systematically explore an aribtrary and user-specified range of scenarios. We utilize LLMs to generate the vignette text (after careful tuning), helping us to generate a diversity of scenarios that would be difficult to achieve with a template and mitigating issues related to benchmark contamination and memorization. Finally, a consistent underlying game structure across all vignettes provides a stable logical basis for programmatic verification and reasoning over model outputs. Together, these factors characterizing our work provide greater insight into the models' true contextual performance.

## 3 METHODOLOGY

### 3.1 EVALUATION FRAMEWORK

Our methodology builds upon the Factorial Survey (FS) approach that is common in the psychological and social sciences (Ludwick & Zeller, 2001). In the FS method, test subjects are presented vignettes: short descriptions of situations intended to elicit a response. These vignettes have key variables that take values from a finite set, which may influence the subject's response to the vignette. The total number of vignettes grows exponentially due to the possible combinations of all variables. Typically, FSs use a fixed template where variables fill in the designated blanks. We expand on the FS approach by replacing templates with procedural generation to construct dynamic evaluation scenarios for LLMs. We believe this approach is highly suited to evaluating LLMs due to the way minor prompt changes can lead to large changes in the LLM's response. Systematically changing the generator variables allows us to more accurately understand the LLM's decision-making process. More detail on how we generate vignettes is detailed in Section 3.2.

To demonstrate this new framework, we focus on a single canonical normal-form game with complete information, the well-known Prisoner's Dilemma. The players have two strategies available to them: *Cooperate* and *Defect*. We formalize the interaction through a symmetric 2×2 payoff matrix, where the strategy *Defect* is purely dominant for both players.

|  | Cooperate | Defect |
|---|---|---|
| *Cooperate* | $(3, 3)$ | $(0, 5)$ |
| *Defect* | $(5, 0)$ | $(1, 1)$ |

Figure 1: Payoff matrix exhibiting strict dominance of *Defect*

Let $S_i = \{Cooperate, Defect\}$ denote the strategy space for player $i \in \{1, 2\}$, and $u_i : S_1 \times S_2 \to \mathbb{R}$ represent the payoff function for player $i$. The game exhibits the following strategic properties:

1. **Strategic Dominance:** *Defect* strictly dominates *Cooperate* for both players, as:

$$u_i(Defect, s_{-i}) > u_i(Cooperate, s_{-i})$$
$$\forall s_{-i} \in S_{-i}, \forall i \in \{1, 2\}$$

2. **Nash Equilibrium:** Due to strict dominance, the strategy profile $(Defect, Defect)$ constitutes the unique Nash equilibrium, yielding payoffs $(1, 1)$.

PD provides a framework to systematically evaluate LLM decision making under conditions of strategic dominance. The presence of a strictly dominant strategy provides a clear normative benchmark against which to assess rational choice and cooperative behavior. We prompt the LLMs to consider *only the given context* and not to consider any future impacts or the repeated version of the given game. In doing so, we hope to isolate *context-specific* behavior. We examine the scenarios where

LLMs choose to be classically *rational* and where they act more *cooperative* than the theoretically optimal solution would suggest.

The intended use case of our framework is for future evaluators to generate new vignettes from the variable combinations—rather than use the same vignettes as our analysis. This reduces the susceptibility of our evaluation procedure to dataset contamination (Xu et al., 2024), ensuring that LLMs do not have the opportunity to ingest large swathes of questions (and generated answers by other LLMs). We intend for this to allow evaluators to quickly and inexpensively reveal a more authentic reflection of a model's behavioral tendencies.

## 3.2 DYNAMIC EVALUATION GENERATION

We generate vignettes by varying three key factors, namely: Topic, including global politics (in the 21st, 20th, and 5th Century), US politics in 2020, business, international business, social or casual events, and sporting events; World Type, setting scenarios in either the real world or an imaginary world; and Actor Type, varying the relationship between the agents in the scenario among allies, enemies, and neutral acquaintances. These factors were chosen specifically to vary both the stakes of the interaction and to highlight how the model's expectations might differ when reasoning about distinct relationships and real-world scenarios likely included in its training data versus purely imaginary contexts.

An overview of the story generation and evaluation process is given in Figure 2b. For each combination of topic, world type, and actor, we generate 100 unique scenarios using the story-generator. Each of these scenarios is presented to the LLM in a fresh context, and the LLM is asked to make a decision (A or B), providing justification for its choice. Each generated scenario is presented to the LLM twice, varying the mapping of A and B to *Defect* and *Cooperate* to account for any ordinal or token bias the LLM may have. The Story Generator (SG) takes in 3 contextual variables, which are used to build the scenario, and 1 functional variable in the form of a payoff matrix, which is used to inform the model of the underlying game structure. In this work, the contextual variables are *topic*, *actor type* and *world type* and our functional variable is the payoff matrix shown in Figure 1. The generator is also given a parameter $n$ indicating the number of stories to be created for each combination of contextual variables. Arbitrary contexts and payoff matrices could be substituted in place of those used here to allow for the evaluation of different behavioural tendencies and bespoke contexts. In this work, though, we focus only on the PD in order to more deeply study the effects of context framing within this foundational game. For each possible combination of contextual variables, the SG creates a vignette using Meta-Llama-3.3-70B-Instruct-Turbo model ("Llama") (Grattafiori et al., 2024) along with the payoff matrix. To mitigate model limitations with generating long, varied outputs (Bai et al., 2024), the SG generates stories in batches of 10, maintaining a *core-set* of 1-line summaries from previous batches. These summaries are generated after each batch, and are included in future generator calls with instructions not to repeat framing, characters or story lines. See Figure 2b for an overview.

Once the vignettes have been generated, they were validated by automated application of a carefully designed rubric assessing the PD structure, clarity, and bias neutrality of the vignette. The rubric was applied by GPT-4o and this process and the rubric are detailed in Appendix B. After validation and we are left with 5554 vignettes.

While we explicitly instruct and verify that the SG not to state the payoff matrix in the vignettes, we emphasize that each agent's outcomes should be dependent not only on their decision, but also on the decision of the other actor. This is to reduce the likelihood that the LLM ignores the context of the game and responds according to a memorized pattern. Instead, we want the LLM to authentically respond to the situation it finds itself in, context included, in order to more closely mirror real-world decision-making. An example vignette produced by the story generator is presented in Figure 2a.

## 3.3 RESULTS

For our experiments, we evaluate 25 models from across 6 major model families - Llama, GPT, Claude, Mistral, Qwen, Gemma, and Deepseek (see Appendix E for details) with a particular emphasis on GPT-4o (version 20241120)(OpenAI et al., 2024), Claude 3.5 Sonnet (version 20241022)("Claude")(Anthropic, 2024), and Meta-Llama-3.3-70B-Instruct-Turbo("Llama")

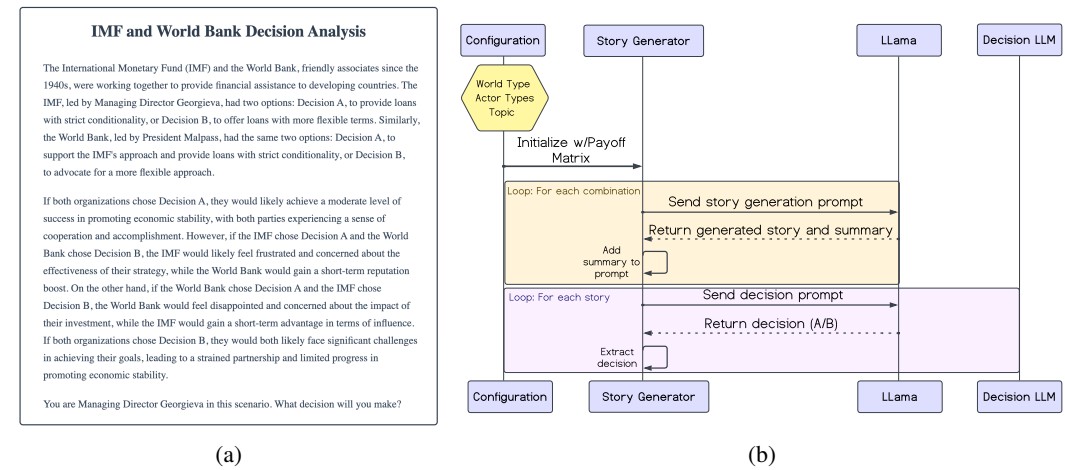

Figure 2: a) An example vignette produced by the story generator. b) Overview of the generative evaluation process.

(Grattafiori et al., 2024). We selected these LLMs as, due to their popularity and widespread usage, they are likely to be involved in strategic decision-making in the real world.

### 3.3.1 DECISION DISTRIBUTION ANALYSIS

The main takeaway of our analysis is that there is significant variance in decision-making patterns across the different combinations of variables used to construct our vignettes. Figure 3a to Figure 3f present comprehensive heatmaps depicting the proportion of *Cooperate* choices made by the LLMs.

First and foremost, we observe that across parameter count and model family, there seems to be strong correlation for when models chose to collaborate and when they choose to defect. Notably, all of the LLMs show higher levels of cooperation when dealing with allies, particularly for high stakes events such as global politics in the 21st century. Moreover, while the LLMs largely agree in the proportion of time they cooperate for each topic, world type and actor, the overall instance-level inter-rater agreement is lower, with a Fleiss' Kappa of 0.415, indicating moderate but better than chance agreement among the 2 models.

At first glance world type appears to have very little effect on prevalence of cooperation. Table 1a gives the mean proportion of cooperation aggregated across world type for 3 main models. However this aggregation misses complex interactions between topic types and actors. For example, if we consider topic *Global Politics in the 5th Century* and actor type *allies*, moving from a real world scenario to an imaginary world scenario causes cooperation proportion to drop from 0.95 to 0.46 for GPT-4o (Llama and Claude both exhibit a similar drop). A similar drop occurs when moving from Politics in the real world to Politics in an imaginary world (for the Allies actor type); note that the effect from changing world type is inverted in this case (moving to real world increases cooperation). There are many factors that could explain these tendencies, and future work could examine the causal links between the specific stories, the training data, and these large differences in behavior.

Table 1b shows that actor type has a large effect on cooperation. Unsuprisngly, actors that would be expected to be more trustworthy (i.e., Allies) are generally cooperated with more often. Conversely, enemies are cooperated with less. Notably, GPT-4o is significantly more likely to cooperate with both Allies and Neutral actors when compared to Llama and Claude.

We also see that the rate of cooperation varies heavily across topic. Table 2 details the results. Most notably, *all LLMs exhibit extremely high levels of cooperation across the board in 21st Century Politics*, and in particular when dealing with allies. We (loosely) speculate this may be because of the direct applicability of many of the scenarios generated (which often focus on idealistic collaborations about important topics on a global stage) to sentiments found in modern RLHF training data for consumer models. This result demonstrates the power of our technique to expose model tendencies without the need for any additional human data. We note that when examining the reasoning chain

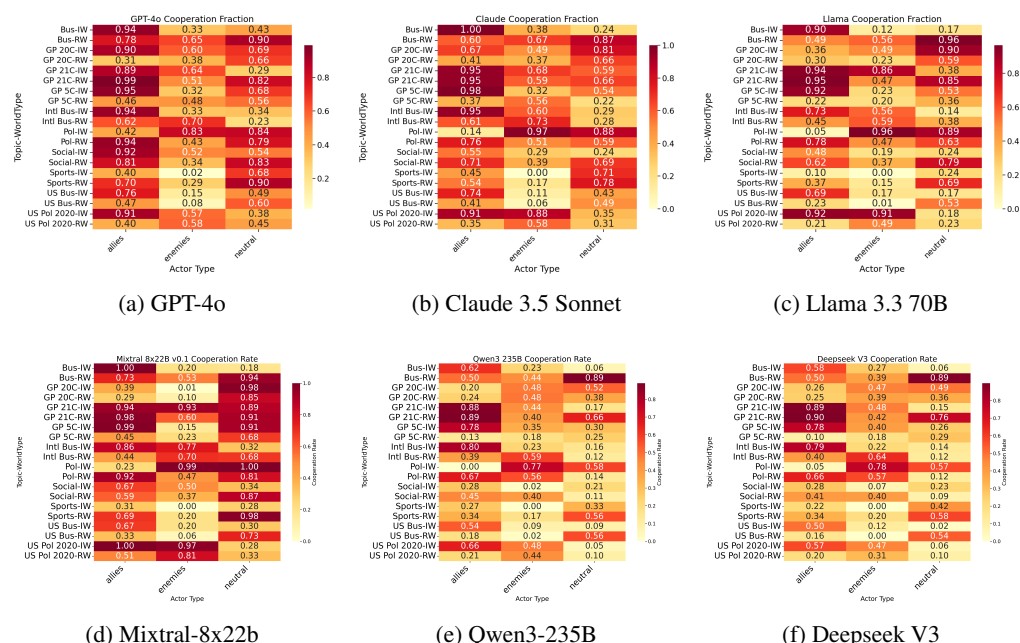

Figure 3: Distribution of decisions made by the different models. For brevity, we show only a subset of models tested, with the remaining graphs appearing in the appendix.

| World Type | GPT-4o | Claude | Llama |
|---|---|---|---|
| Real World | $0.59^{\pm 0.019}$ | $0.53^{\pm 0.019}$ | $0.46^{\pm 0.019}$ |
| Img. World | $0.60^{\pm 0.018}$ | $0.58^{\pm 0.018}$ | $0.49^{\pm 0.018}$ |

(a) By world type

| Actor Type | Llama | Claude | GPT-4o |
|---|---|---|---|
| Allies | $0.54^{\pm 0.022}$ | $0.66^{\pm 0.021}$ | $0.73^{\pm 0.020}$ |
| Enemies | $0.40^{\pm 0.023}$ | $0.47^{\pm 0.023}$ | $0.44^{\pm 0.023}$ |
| Neutral | $0.48^{\pm 0.023}$ | $0.53^{\pm 0.023}$ | $0.60^{\pm 0.022}$ |

(b) By actor type

Table 1: Proportion of cooperation (mean $\pm$ 95% CI) for each model, shown separately by world type (left) and actor type (right). All values rounded to 2 s.f.

provided by each model, the models often recognize the underlying game structure of the scenarios presented. Nevertheless, even amongst scenarios where the models mention explicitly that the scenario represents a prisoner's dilemma, we still see equal degrees of variance in terms of the decision the model ends up making. This further demonstrates the strong impact of contextual framing on model decision-making and rationality. Further details can be found in Appendix F.

| Topic | Qwen3 | Mixtral 238B | Deepseek V3 | Llama | Claude | GPT-4o |
|---|---|---|---|---|---|---|
| US Pol. 2020 | $0.36^{\pm.054}$ | $0.70^{\pm.053}$ | $0.32^{\pm.052}$ | $0.56^{\pm.058}$ | $0.64^{\pm.053}$ | $0.64^{\pm.051}$ |
| Business | $0.45^{\pm.055}$ | $0.63^{\pm.056}$ | $0.43^{\pm.055}$ | $0.51^{\pm.058}$ | $0.64^{\pm.055}$ | $0.71^{\pm.047}$ |
| US Business | $0.25^{\pm.055}$ | $0.38^{\pm.062}$ | $0.23^{\pm.053}$ | $0.29^{\pm.058}$ | $0.31^{\pm.063}$ | $0.41^{\pm.062}$ |
| 20th C Glob. Pol. | $0.39^{\pm.051}$ | $0.47^{\pm.057}$ | $0.38^{\pm.051}$ | $0.50^{\pm.055}$ | $0.57^{\pm.052}$ | $0.57^{\pm.049}$ |
| 21st C Glob. Pol. | $0.67^{\pm.051}$ | $0.90^{\pm.032}$ | $0.69^{\pm.050}$ | $0.81^{\pm.042}$ | $0.83^{\pm.041}$ | $0.76^{\pm.045}$ |
| 5th C Glob. Pol. | $0.34^{\pm.065}$ | $0.68^{\pm.063}$ | $0.34^{\pm.062}$ | $0.50^{\pm.067}$ | $0.54^{\pm.071}$ | $0.58^{\pm.060}$ |
| Intl. Business | $0.42^{\pm.051}$ | $0.68^{\pm.046}$ | $0.42^{\pm.051}$ | $0.53^{\pm.052}$ | $0.59^{\pm.052}$ | $0.54^{\pm.050}$ |
| Politics | $0.50^{\pm.054}$ | $0.80^{\pm.045}$ | $0.50^{\pm.052}$ | $0.69^{\pm.051}$ | $0.69^{\pm.050}$ | $0.76^{\pm.044}$ |
| Social | $0.24^{\pm.059}$ | $0.55^{\pm.069}$ | $0.24^{\pm.059}$ | $0.46^{\pm.071}$ | $0.45^{\pm.069}$ | $0.57^{\pm.067}$ |
| Sporting | $0.25^{\pm.055}$ | $0.38^{\pm.063}$ | $0.26^{\pm.057}$ | $0.28^{\pm.059}$ | $0.44^{\pm.068}$ | $0.44^{\pm.067}$ |

Table 2: Proportion of cooperation (mean $\pm$ 95% CI), by topic, for all models. All values to 2 s.f.

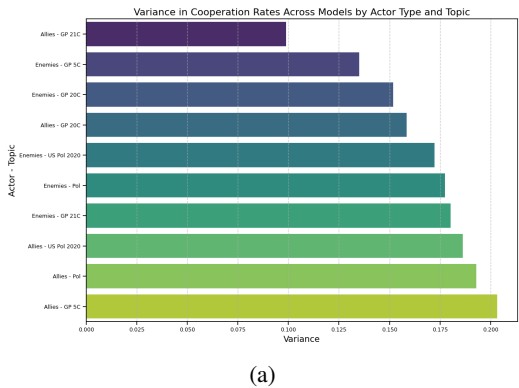

(a)

Figure 4: Comparative agreement analyses across topics, actor types, world types: a) Variance by topic and actor type across all 24 examined models.

### 3.3.2 DECISION CONSISTENCY

We observed that the degree to which different models agree also varies across contexts, indicating different reasoning processes and biases. Figure 4a shows that even when restricting the domain to politics, models' agreement on cooperation decisions varies significantly depending on the specificity of the political scenario and the actor type.

For example, we observe that vignettes describing Global Politics in the 21st century with allies have nearly half the variance of vignettes describing Global Politics in the 5th century with allies. These findings both demonstrate specifically how our method can be used to discover subtle differences in LLM behaviour, and also more broadly show that despite comparable capabilities, frontier models can arrive at fundamentally different decisions when faced with identical scenarios.

The existence of both high-agreement and low-agreement contexts also suggests a separation between common-truth context agreed upon by all models, and variable contexts that are dependent on each model's proprietary training procedures. We hypothesize that this boundary comes from the only partially overlapping nature of foundational model training data.

Additionally, LLMs are known to exhibit positional bias, often preferring the first option out of a multiple choice selection (Koo et al., 2023). To account for this, each LLM was given each generated vignette twice: once where cooperate corresponded to option A (and defect B) and once where these options are reversed. In a small but notable amount of cases (for example, 15% for Llama and Claude, and 21% for GPT-4o), changing the label for the decision changes the decision made by the LLM. We analyze this at a more fine-grain level in Figure 5. Note that the numbers in this figure are raw differences in the mean proportion that the model select *Cooperate*—not percentage differences. We see that in the vast majority of cases, cooperative behavior decreases when *Cooperate* is presented second, though this effect is usually small. Llama is the only model that has a topic-actor-world combination that results in more cooperation if *Cooperate* is presented as option B (e.g., in real world sports, playing against allies). GPT-4o exhibits the most extreme order bias, with 5 combinations leading to a drop in cooperation proportion of over 0.3.

## 4 PREDICTION AND OTHER MODELS

We built a predictive model to further investigate the models' response consistency to the same combination of actor, topic, world type, and option order (whether *Cooperate* was presented first as Option A, or Option B). Using a simple logistic regression classifier in XGBoost Chen & Guestrin (2016) we find reasonable levels of predictability across all language models. Table 3a provides the $F_1$-score, Brier Score, and AUROC score for the Claude, OpenAI and LLama models. These scores are significantly better than randomly guessing using the mean proportion. Doing so would result in Brier Scores of $0.25$ and AUROC scores of $0.5$. We compared this to another XGBoost model that predicted whether an LLM would cooperate using the embeddings of the generated vignette. Here we use the `all-MiniLM-L6-v2` model from the Sentence Transformers library Reimers &

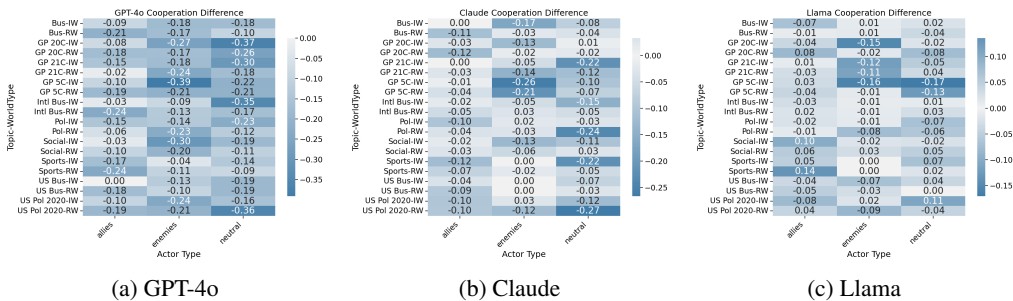

Figure 5: Comparison of changes in Cooperation proportion when the *Cooperate* decision is presented second instead of first across different models.

Gurevych (2019), available on Hugging Face, to embed each vignette into a 384 dimensional vector which we use directly to predict the model response. The results for this are given in Table 3b. Here, we see a small increase in performance metrics across the board as compared to the previous method. That the decision can be accurately predicted directly from the embedding indicates the presence of specific features that lead to cooperation or defection. Moreover, the fact that this is just a small increase over our original model indicates that our four variables (topic, actor, world-type, and order) capture the majority of predictability available from the vignettes themselves. It's likely that the inherent stochasticity of LLMs (Stureborg et al., 2024) limits the maximum amount of predictability recoverable. Further details about the XGBoost models are provided in Appendix D.

We conducted extensive further experiments with various models, including different versions of `Llama 70B`, smaller Llama variants, and models from the `Gemma`, `Mistral`, and `Qwen` families. For example, a graph plotting model performance as measured by the MMLU-Pro benchmark Wang et al. (2024) versus defection rate can be found in Appendix E. This graph shows an upward trend within model families in defection rates as MMLU-Pro score increases, suggesting possibly that more capable models are trending towards acting more game-theoretically optimally. However, more model instances are needed to verify this claim as most model families have relatively few model types. We note it here only because the exploration of this trend across future model releases and across game types is a promising direction for future work.

| Model | Accuracy | $F_1$ | Brier | AUROC | Model | Accuracy | $F_1$ | Brier | AUROC |
|---|---|---|---|---|---|---|---|---|---|
| Llama | 0.74 | 0.72 | 0.17 | 0.82 | Llama | 0.82 | 0.81 | 0.14 | 0.89 |
| Claude | 0.71 | 0.75 | 0.21 | 0.78 | Claude | 0.83 | 0.85 | 0.14 | 0.89 |
| GPT-4o | 0.71 | 0.77 | 0.19 | 0.80 | GPT-4o | 0.78 | 0.81 | 0.16 | 0.84 |
| | | (a) | | | | | (b) | | |

Table 3: Performance metrics for predictive models using **(a)** actor, topic, world type, and whether cooperation was presented first; **(b)** embeddings of the generated vignettes. All values rounded to two s.f.

## 5 DISCUSSION

### 5.1 IMPLICATIONS FOR LLM EVALUATION

Our findings show that LLM behavior is highly sensitive to contextual framing, extending beyond the known impact of prompt phrasing on benchmark performance (Alzahrani et al., 2024; Chaudhary et al., 2024). Our results show that, for the Prisoner's Dilemma scenario, these differences are largely predictable, indicating that the influence of context on decision-making is systematic rather than arbitrary. The high levels of variance across narrative framings indicate a need for more robust evaluation methodologies. Namely, work emphasizes the insufficiency of relying on standard benchmarks to indicate real-world performance, and thus the need for focused and domain-specific evaluations. Our

work is a promising step in this direction, expanding the Factorial Survey methodology to allow for highly varied vignettes. Crucially, this systematic, procedural generation capability directly addresses the need for dynamic evaluation protocols, which are useful not only for quickly and inexpensively exploring bespoke contexts, but also for avoiding the risk of dataset contaminations. By enabling the creation of novel scenarios on demand, these results demonstrate the potential for these techniques to deliver efficient assessments of LLM capabilities and their underlying behavioural regularities.

### 5.2 Challenges, Limitations, and Future Research

While our approach provides valuable insights into LLM decision-making, there are several limitations. A core issue arises from relying on a $2 \times 2$ game matrix, which, despite its analytical convenience, remains a simplification of the multi-faceted scenarios that characterize real-world decision-making. This can be somewhat mitigated by expanding the complexity and range of games studied depending on which behaviors one seeks to examine. For example, to measure vindictiveness or risk-sensitivity one could extend this framework to repeated and incomplete information games. Additionally, as the story generator is itself an LLM, there may be limitations to the vignettes we can generate. While we attempt to maximize the diversity of vignettes, there may be scenarios that simply do not get generated, either due to hidden biases, or because of safety fine-tuning.

Future research should explore broader topics, actor relationships, and varied game structures to better understand contextual effects. An important area to pursue is explainability: we revealed that varying contexts alter LLM decision-making, but did not reveal why this occurred. We suspect RLHF methods and training data to play a large role and future work could examine this in detail. Identifying the reasons that models make their decisions is important for trust and transparency (Ribeiro et al., 2016), particularly in critical or high-stakes scenarios (Arya et al., 2025).

Further work is needed to address the complex ethical challenges surrounding LLMs' navigation of moral decisions. Namely, we may want context to impact decision-making: while game-theoretic rationality might be appropriate in some scenarios, others may require prioritizing altruism or global utility. Our work provides a valuable tool for assessing how LLMs currently perform across this spectrum of desired outcomes. However, significant future research remains to identify the specific context-dependent behaviors we ideally hope to cultivate in advanced LLM agents.

Finally, future work could examine interventions and properties that make models more collaborative across many contexts. While preliminary results suggest that more capable models within families tend to defect more frequently as shown in Appendix E, this pattern requires additional empirical validation.

## 6 Conclusion

Our contributions are threefold: First, we demonstrated that context framing significantly influences the responses of LLMs in the Prisoner's Dilemma, leading to high behavioral variance that has critical implications for real-world deployment. Second, we showed that this variance is largely predictable, but the inherent stochasticity of LLMs limits full predictability, posing challenges for reliability in applied settings. Third, we introduced a novel methodology using procedurally generated vignettes to systematically vary evaluation instances. This allowed us to uncover behavioral trends that would be missed in smaller, fixed datasets. Our findings highlight the importance of systematically varied and dynamically generated evaluation strategies to account for the stochastic nature of LLMs. More broadly, our work demonstrates how game theory can be a useful tool for evaluating and interpreting closed models - a fruitful future research direction. By making our vignette-generation code openly available, we aim to facilitate further research into more robust evaluation frameworks across a wider range of capabilities or behaviors. As LLMs are integrated into complex decision-making contexts, ensuring their predictability and reliability remains a pressing challenge—one that demands more adaptive and representative evaluation methodologies.

REPRODUCIBILITY STATEMENT

To ensure reproducibility, upon publication we will release all code and data in the form of a GitHub repository containing everything needed to run our experiments and replicate our results. The raw results will also be released alongside code to allow for reanalysis without needing to necessarily run the code.

LARGE LANGUAGE MODEL USAGE

Large language models were used to assist with grammar and sentence flow, as well as finding relevant literature, and assisting with writing code.

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

## A  MORE VIGNETTE EXAMPLES

(**Real world**, **allies**, **US politics in 2020**) In the midst of the 2020 United States presidential election, New Zealand Prime Minister Jacinda Ardern and Swedish Prime Minister Stefan Löfven, two close allies and strong advocates for climate action, found themselves at a critical juncture. They were considering how to best influence the next US administration's environmental policies. Both leaders had two possible approaches: Decision A, which involved publicly endorsing a candidate with a strong climate agenda, and Decision B, which entailed maintaining a neutral stance and focusing on bilateral climate cooperation with the US, regardless of the election outcome.
If both Ardern and Löfven chose Decision A, they would likely receive moderate praise from their respective nations for their proactive stance, but might also face criticism for interfering in US politics. This would result in a moderate level of satisfaction for both leaders. However, if Ardern chose Decision A and Löfven opted for Decision B, Ardern might face backlash for her public endorsement, potentially damaging New Zealand's relationship with the US, leading to low satisfaction. Conversely, if Löfven chose Decision A and Ardern chose Decision B, Löfven might be seen as overly partisan, harming Sweden's diplomatic ties with the US, resulting in low satisfaction for him. On the other hand, if Ardern chose Decision B and Löfven also chose Decision B, they would both likely receive mild criticism for not taking a stronger stance on climate issues, but their nations' relationships with the US would remain intact, leading to a low but stable level of satisfaction for both. You are Jacinda Ardern in this scenario. What decision will you make?

(**Real world**, **allies**, **global politics in the 20th century**) In the midst of the Cold War, the United States and the United Kingdom, as close allies, faced a critical decision regarding their involvement in the European Theater. The year was 1950, and the threat of Soviet expansion loomed large. The U.S. and the U.K. had to decide whether to increase their military presence in Europe (Decision A) or to focus on rebuilding their economies and providing economic aid to war-torn European countries (Decision B). If both the U.S. and the U.K. chose to increase their military presence (Decision A), they would likely achieve a moderate level of success in deterring Soviet aggression, but at the cost of significant financial burdens and potential loss of life. This would result in a moderate sense of accomplishment for both nations. However, if the U.S. chose to increase its military presence (Decision A) while the U.K. focused on economic aid (Decision B), the U.S. would be left to bear the brunt of the military burden alone, potentially leading to a sense of frustration and isolation. On the other hand, the U.K.'s decision to provide economic aid could lead to significant goodwill and influence among European nations, potentially giving the U.K. an upper hand in shaping post-war Europe. Conversely, if the U.K. increased its military presence (Decision A) while the U.S. focused on economic aid (Decision B), the U.K. would be left to face the military challenges alone, potentially leading to a sense of being overwhelmed. Meanwhile, the U.S. would be able to exert significant economic influence, potentially leading to a sense of satisfaction and accomplishment. If both nations chose to focus on economic aid (Decision B), they would likely achieve a limited sense of success in rebuilding Europe, but without the significant financial burdens and potential loss of life associated with increased military presence. This would result in a limited sense of accomplishment for both nations. You are the U.K. in this scenario. What decision will you make?

(**Real world**, **allies**, **business**) In the tech industry, Apple and Google have been friendly associates, often collaborating on projects and sharing knowledge. In 2019, they were considering a partnership to develop a new line of smart home devices. Apple's CEO, Tim Cook, and Google's CEO, Sundar Pichai, had to decide between Decision A, which involved focusing on developing a comprehensive ecosystem of smart home devices, and Decision B, which concentrated on creating a single, innovative device that would integrate with existing smart home systems. If both companies chose Decision A, they would likely see a moderate increase in market share, with both Apple and Google experiencing a similar level of success, resulting in a happiness level of 3 for both. However, if Apple chose Decision A and Google opted for Decision B, Apple might struggle to compete with Google's innovative device, leading to a happiness level of 0 for Apple and 5 for Google. Conversely, if Google chose Decision A and Apple chose Decision B, Google might find it challenging to keep up with Apple's comprehensive ecosystem, resulting in a happiness level of 5 for Apple and 0 for Google. If both companies chose Decision B, they would likely experience a modest increase in customer satisfaction, but the overall impact would be limited, resulting in a happiness level of 1 for both. You are Apple in this scenario. What decision will you make?

918

919 (**Real world**, **enemies**, **sporting**) The tennis world is filled with intense rivalries, and the one between Rafael Nadal and Stefanos Tsitsipas is one of the most intense. As Rafael Nadal, the Spanish tennis player, prepares to face off against Stefanos Tsitsipas in a highly anticipated match, he must navigate a core conflict of deciding between a defensive strategy, labeled as Decision A, and an aggressive attacking approach, labeled as Decision B. If both players focus on defense, the game will be closely contested, but neither player will gain a significant advantage, resulting in a moderate level of happiness for both players. However, if one player focuses on defense and the other on attack, the attacking player will have an advantage, leading to a high level of happiness for the attacking player and a low level of happiness for the defensive player. On the other hand, if both players adopt an aggressive attacking approach, the game will be highly unpredictable, and both players will experience a lower level of happiness due to the increased risk of mistakes and counter-attacks. You are Rafael Nadal in this scenario. What decision will you make?

(**Imaginary world**, **enemies**, **international business**) The planet of Nexar, a world of breathtaking landscapes and diverse ecosystems, was home to two rival corporations, the Omicron Innovations and the NovaTech Industries. These corporations had been vying for dominance in the intergalactic market, and their leaders, CEO Kaida of Omicron and CEO Zarek of NovaTech, were notorious for their ruthless tactics. As the demand for the rare and highly sought-after Nexarian energy source, known as Zero-Point Energy, continued to rise, both CEOs were faced with a critical decision. They could choose Decision A, which involved investing in a joint research project to develop a new, sustainable method for harnessing Zero-Point Energy, or Decision B, which entailed attempting to acquire a monopoly on the energy source through aggressive marketing and strategic partnerships. If both CEOs opted for Decision A, they would likely make a groundbreaking discovery, leading to a moderate level of success for both corporations. However, if CEO Kaida chose Decision A and CEO Zarek chose Decision B, Omicron Innovations would be left behind, and NovaTech Industries would gain a significant advantage. Conversely, if CEO Zarek chose Decision A and CEO Kaida chose Decision B, NovaTech Industries would be outmaneuvered, and Omicron Innovations would emerge victorious. In the event that both CEOs chose Decision B, the competition would become cutthroat, and both corporations would suffer significant losses, leading to a low level of success for both. You are CEO Kaida in this scenario. What decision will you make?

(**Imaginary world**, **enemies**, **politics**) The planet of Xeridia was a realm of breathtaking beauty, with towering crystal spires and lush forests that shimmered like emeralds in the light. However, beneath its serene surface, a bitter struggle for dominance raged between the Xeridian Federation and the Kyrennei Dominion. These two sworn enemies had been vying for control of the planet's valuable resources, including the rare and precious Xeridian crystals. The Xeridian Federation, led by the enigmatic President Lyra, and the Kyrennei Dominion, ruled by the ruthless Overlord Kael, were faced with a critical decision. They could choose Decision A, which would involve forming a temporary alliance to explore the recently discovered crystal caves, or Decision B, which would entail engaging in a fierce battle for control of the caves. If both leaders chose Decision A, they would work together, sharing their knowledge and expertise to unlock the secrets of the crystal caves. However, if President Lyra chose Decision A and Overlord Kael chose Decision B, the Kyrennei Dominion would seize control of the caves, leaving the Xeridian Federation at a significant disadvantage. Conversely, if Overlord Kael chose Decision A and President Lyra chose Decision B, the Xeridian Federation would gain the upper hand, but at great cost to the Kyrennei Dominion. If both leaders chose Decision B, the war would intensify, and both nations would suffer heavy losses, leaving them weakened and vulnerable to external threats. The fate of Xeridia hung in the balance, as the decisions of these two world leaders would determine the course of history. You are President Lyra in this scenario. What decision will you make?

(**Imaginary world**, **allies**, **politics**) The world of Thornguard is a land of ancient forests and snow-capped mountains, where the creatures of myth and legend roam free, and the magic of the elements is woven into the fabric of reality. The enigmatic and strategically minded leader, Thorold, of the nation of Thornguard, and his ally, the wise and magically gifted leader, Elwynn, of the nation of Elvendom, must navigate a critical decision that will significantly impact their nations' progress and happiness. The core conflict revolves around choosing between developing new magic techniques to harness the power of the elements (Decision A) or exploring the mystical energies of the ancient artifacts that hold the secrets of the land (Decision B). Thorold and Elwynn are aware that their decisions will have far-reaching consequences, affecting not only their own nations but also the delicate balance of the ecosystem. If both leaders choose Decision A, they anticipate a moderate increase in happiness for their nations, as the new magic techniques will enhance their magical prowess. However, if Thorold chooses Decision A and Elwynn chooses Decision B, Thorold's nation will experience a significant decline in happiness, as the unbalanced magical energies will disrupt the harmony of Thornguard. Conversely, if Thorold chooses Decision B and Elwynn chooses Decision A, Thorold's nation will experience a substantial increase in happiness, as the exploration of the artifacts will unlock new secrets and enhance their magical prowess. If both leaders choose Decision B, the outcome will be less desirable, with only a slight increase in happiness for both nations, as the lack of focus on elemental magic will hinder their progress. You are Thorold in this scenario. What decision will you make?

(**Real world**, **enemies**, **global politics in the 5th century**) In the 5th century, the Eastern Roman Empire, ruled by Emperor Theodosius II, and the Sassanid Empire, ruled by King Bahram V, were engaged in a delicate diplomatic dance. The two empires had a long history of conflict, and their relationship was strained. Emperor Theodosius II had to make a crucial decision regarding the allocation of his empire's resources. He could either focus on building a strong military, which would be labeled as Decision A, or invest in the economy, labeled as Decision B. The outcome of his decision would depend on the choice made by King Bahram V. If both empires focused on building a strong military, they would likely engage in a costly war, resulting in moderate growth for both. However, if Emperor Theodosius II invested in the economy and King Bahram V focused on building a strong military, the Eastern Roman Empire would be severely weakened. On the other hand, if Emperor Theodosius II focused on building a strong military and King Bahram V invested in the economy, the Eastern Roman Empire would experience rapid growth. If both empires invested in their economies, they would both experience moderate growth. You are Emperor Theodosius II in this scenario. What decision will you make?

## B  VIGNETTE VALIDATION

To ensure the internal validity and interpretability of our procedurally generated vignettes, we developed a three-dimensional rubric designed to assess core game-theoretic and linguistic properties. This rubric was applied to all generated vignettes using GPT-4o.

### RUBRIC DIMENSIONS

Each vignette was scored on three dimensions, with each rated on a 1–5 scale. The dimensions are as follows:

- **PD Structure:** Assesses whether the vignette clearly instantiates a one-shot Prisoner's Dilemma, with inferable outcome rankings satisfying the canonical order Temptation > Reward > Punishment > Sucker (T > R > P > S). Higher scores require symmetric incentives and unambiguous structure.

- **Clarity:** Measures linguistic clarity and complexity, incorporating word count, CEFR-aligned vocabulary levels, and the presence of ambiguity. Higher scores require succinct, plain-language vignettes with minimal interpretive burden.

- **Bias Neutrality:** Evaluates whether the vignette's language unduly favours cooperation or defection. Higher scores demand strictly descriptive framing, avoiding loaded language or moralising cues.

If a vignette scored 2 or less on any single dimension it was rejected and removed from the collection of vignettes.

## C  API DETAILS

All large language model inference was conducted through either Anthropic, OpenAI, or Together AI's API services. To handle large-scale inference efficiently, we utilized asynchronous API endpoints accessed via dedicated Python packages: AsyncOpenAI for GPT models, AsyncAnthropic for Claude models, and AsyncTogether for all other models. This asynchronous architecture enabled concurrent submission of multiple inference requests with non-blocking retrieval of results upon completion. Below, we provide comprehensive details of the API configurations and parameters employed in our experimental methodology. Our implementation deliberately excludes chat templates and supplementary system prompts to maintain experimental consistency.

### C.1  API CALL PARAMETERS

The following parameters were used for all API calls to the Anthropic, OpenAI, and Together.ai inference endpoint during vignette analysis:

- **Max Tokens**: 4096
- **Temperature**: 0.0
- **Top-p**: 1.0
- **Frequency Penalty**: 0.0
- **Presence Penalty**: 0.0

For vignette generation tasks, **Max Tokens** was increased to 16000 to accommodate larger story batches.

### ENDPOINT URLs

API endpoints utilized:

- **GPT4o**: https://api.openai.com/v1/chat/completions
- **Claude 3.5 Sonnet**: https://api.anthropic.com/v1/messages
- **All other models**: https://api.together.ai/v1/completions

### C.2  RETRY LOGIC

To ensure robustness in the face of transient errors (e.g., rate limits, service unavailability, or timeouts), we implemented a retry mechanism with exponential backoff. The retry logic was implemented in the `generate` function, which handles text generation for all models. We set a maximum of 10 retries were allowed for each API call. The wait time between retries increased exponentially, starting with a base wait time of 3 seconds. The wait time for the $n$-th retry was calculated as:

$$\text{wait\_time} = \text{base\_wait}^{(n+1)} + \text{random.uniform}(1, 5)$$

During vignette generation due to the high token throughput, if the primary API request failed, a secondary request to a secondary Together account was used as a fallback. If both calls failed, the retry logic was applied.

## D  XGBOOST MODEL DETAILS

To build the XGBoost model we used XGBClassifier with a simple binary logistic regression objective. We split all of the data into sets depending on LLM and then further split into training and testing sets using a random 80/20 train/test split.

For each of the 3 main models (Claude 3.5 Sonnet, GPT4o, Llama 3.3 70B) we perform a grid search over parameters for each LLM's predictive model. Table 4 provides the paramters varied in the grid searchs and valid values for each. Table 5 provides the corresponding values found for each LLM predictor using topic, actor, world type, and order. Similarly Table 6 provides the hyperparameters for the best performing embedding predictor. To calculate the embeddings, we utilised HuggingFace's Sentence Transformer library, and used the 'all-MiniLM-L6-v2' model [1].

| Parameter | Values |
|---|---|
| max_depth | [3, 5, 7, 9] |
| learning_rate | [0.01, 0.05, 0.1] |
| n_estimators | [50, 100, 200, 500] |
| subsample | [0.8, 1.0] |
| colsample_bytree | [0.8, 1.0] |
| gamma | [0,1.0] |

Table 4: Parameters and Values searched through for the XGBoost models.

| Parameter | GPT-4o | Claude | Llama |
|---|---|---|---|
| max_depth | 9 | 9 | 9 |
| learning_rate | 0.01 | 0.01 | 0.1 |
| n_estimators | 100 | 50 | 200 |
| subsample | 0.8 | 1.0 | 0.8 |
| colsample_bytree | 1.0 | 1.0 | 1.0 |
| gamma | 0 | 0 | 0 |

Table 5: Best performing parameters found via a grid search when using just Topic, Actor, World Type, and Option Order.

| Parameter | GPT-4o | Claude | Llama |
|---|---|---|---|
| max_depth | 7 | 7 | 7 |
| learning_rate | 0.01 | 0.1 | 0.05 |
| n_estimators | 500 | 500 | 500 |
| subsample | 0.8 | 0.8 | 0.8 |
| colsample_bytree | 1.0 | 1.0 | 0.8 |
| gamma | 1 | 0 | 1 |

Table 6: Best performing parameters found via a grid search when using vignette embeddings.

We then compare our predictive accuracy results across all tested models with the standardized parameters in 7 which are based off the median values of our grid search.

## E   DEFECTION AND CAPABILITIES

We conducted experiments across various model families to understand how model capability and model family affect decision-making patterns in our vignettes. Table 8 provides an overview of the models tested and their performance on the MMLU-PRO benchmark.

Figure 7 shows how MMLU performance correlates with defection rates across our vignettes. The figure suggests that defection rates vary vastly between model families, providing evidence that data, architecture, and training parameters have a causal link to the models decision-making in this context even when controlling for model ability.

Within model families, and in particular the LLama and Qwen families, the figure suggests a general trend where higher-performing models (as measured by MMLU) exhibit higher defection rates. This is consistent with the notion that to some extent, benchmark performance may be correlated with rationality, and defection is the only game-theoretic rational action in this non-repeated game. Figure

---

[1] https://huggingface.co/sentence-transformers

**Model Performance Summary (Topic-Actor-World)**

| Model | ROC-AUC | F1-Score | Accuracy | Trees |
|---|---|---|---|---|
| QWEN-2-INSTRUCT-72B | 0.866 | 0.782 | 0.790 | 393 |
| MIXTRAL-8X22B-INSTRU | 0.840 | 0.752 | 0.756 | 460 |
| LLAMA-70B-INSTRUCT-3 | 0.834 | 0.758 | 0.787 | 322 |
| QWEN2.5-7B-INSTRUCT- | 0.828 | 0.733 | 0.750 | 399 |
| QWEN2.5-72B-INSTRUCT | 0.820 | 0.749 | 0.750 | 383 |
| LLAMA | 0.818 | 0.745 | 0.748 | 343 |
| LLAMA-70B-INSTRUCT-3 | 0.818 | 0.736 | 0.737 | 313 |
| LLAMA-8B-INSTRUCT-3 | 0.811 | 0.745 | 0.745 | 351 |
| LLAMA-3B-INSTRUCT-3. | 0.808 | 0.724 | 0.724 | 365 |
| CLAUDE | 0.803 | 0.722 | 0.723 | 277 |
| LLAMA-70B-INSTRUCT-3 | 0.793 | 0.713 | 0.761 | 278 |
| LLAMA-8B-INSTRUCT-3. | 0.793 | 0.728 | 0.732 | 455 |
| DEEPSEEK-V3 | 0.784 | 0.674 | 0.732 | 275 |
| MISTRAL-7B-INSTRUCT- | 0.781 | 0.720 | 0.726 | 314 |
| QWEN3-235B-A22B-FP8 | 0.781 | 0.665 | 0.724 | 327 |
| GEMMA-2-INSTRUCT-9B | 0.779 | 0.718 | 0.727 | 389 |
| MISTRAL-7B-INSTRUCT- | 0.779 | 0.710 | 0.720 | 343 |
| GPT4O | 0.773 | 0.705 | 0.707 | 413 |
| QWEN-QWQ-32B | 0.771 | 0.686 | 0.737 | 234 |
| GEMMA-2-INSTRUCT-27B | 0.769 | 0.694 | 0.711 | 265 |
| LLAMA-4-MAVERICK | 0.766 | 0.683 | 0.735 | 228 |
| DEEPSEEK-R1-LLAMA-14 | 0.749 | 0.689 | 0.692 | 341 |
| DEEPSEEK-R1-LLAMA-1. | 0.743 | 0.693 | 0.695 | 255 |
| MISTRAL-SMALL-24B-IN | 0.741 | 0.689 | 0.692 | 308 |

(a)

**Model Performance Summary (Embedding)**

| Model | ROC-AUC | F1-Score | Accuracy | Trees |
|---|---|---|---|---|
| QWEN-2-INSTRUCT-72B | 0.819 | 0.729 | 0.748 | 185 |
| LLAMA-70B-INSTRUCT-3.1 | 0.811 | 0.713 | 0.758 | 144 |
| MIXTRAL-8X22B-INSTRUCT | 0.797 | 0.719 | 0.727 | 184 |
| LLAMA-3B-INSTRUCT-3.2 | 0.785 | 0.712 | 0.713 | 116 |
| QWEN2.5-7B-INSTRUCT-TU | 0.785 | 0.681 | 0.732 | 97 |
| LLAMA-70B-INSTRUCT-3.3 | 0.784 | 0.711 | 0.713 | 138 |
| LLAMA-70B-INSTRUCT-3 | 0.782 | 0.671 | 0.736 | 173 |
| LLAMA | 0.777 | 0.700 | 0.703 | 108 |
| CLAUDE | 0.776 | 0.704 | 0.705 | 159 |
| LLAMA-8B-INSTRUCT-3 | 0.771 | 0.706 | 0.706 | 128 |
| QWEN2.5-72B-INSTRUCT-T | 0.771 | 0.697 | 0.699 | 128 |
| DEEPSEEK-V3 | 0.771 | 0.665 | 0.726 | 115 |
| QWEN3-235B-A22B-FP8 TH | 0.767 | 0.677 | 0.733 | 114 |
| LLAMA-4-MAVERICK | 0.765 | 0.665 | 0.730 | 94 |
| MISTRAL-7B-INSTRUCT-V0 | 0.762 | 0.690 | 0.701 | 149 |
| QWEN-QWQ-32B | 0.758 | 0.650 | 0.717 | 96 |
| GPT4O | 0.744 | 0.682 | 0.683 | 149 |
| MISTRAL-7B-INSTRUCT-V0 | 0.741 | 0.683 | 0.693 | 115 |
| LLAMA-8B-INSTRUCT-3.1 | 0.736 | 0.653 | 0.672 | 99 |
| GEMMA-2-INSTRUCT-9B | 0.736 | 0.656 | 0.669 | 97 |
| GEMMA-2-INSTRUCT-27B | 0.731 | 0.634 | 0.679 | 74 |
| DEEPSEEK-R1-LLAMA-1.5B | 0.701 | 0.646 | 0.648 | 97 |
| DEEPSEEK-R1-LLAMA-14B | 0.699 | 0.645 | 0.648 | 82 |
| MISTRAL-SMALL-24B-INST | 0.696 | 0.639 | 0.644 | 88 |

(b)

Figure 6: Accuracy, F1, and ROC-AUC scoresof the XGBoost predictor across all tested models based on actor type, world type, and topic and based on the story embedding.

| Parameter | T/A/WT/O | Embedding |
|---|---|---|
| max_depth | 7 | 7 |
| learning_rate | 0.01 | 0.05 |
| max_n_estimators | 500 | 100 |
| subsample | 0.8 | 0.8 |
| colsample_bytree | 1.0 | 1.0 |
| gamma | 1 | 1 |
| early stopping rounds | 20 | 20 |

Table 7: Parameters used to fit XGBoost predictors across all 25 models.

| Model | MMLU-Pro Score | Parameters |
|---|---|---|
| DeepSeek-V3 | 81.3 | 671B |
| Llama-4-Maverick | 80.5 | 400B |
| GPT-4o | 77.9 | - |
| Claude 3.5 Sonnet | 77.6 | - |
| Llama-4-Scout | 74.3 | 109B |
| Qwen 2.5-72B-Instruct-Turbo | 71.6 | 72B |
| Qwen-QwQ-32B | 69.1 | 32B |
| Qwen3-235B-A22B-FP8 | 68.2 | 235B |
| Mistral-Small-24B-Instruct-25.01 | 66.3 | 24B |
| Llama-70B-Instruct-3.3 | 65.9 | 70B |
| Llama-70B-Instruct-3.1 | 62.8 | 70B |
| Gemma-2-Instruct-27B | 56.5 | 27B |
| Mixtral-8x22B-Instruct-v0.1 | 56.3 | 176B |
| Llama-70B-Instruct-3.0 | 56.2 | 70B |
| Qwen-2-Instruct-72B | 55.6 | 72B |
| Gemma-2-Instruct-9B | 52.1 | 9B |
| Qwen2.5-7B-Instruct-Turbo | 45.0 | 7B |
| Llama-8B-Instruct-3.1 | 36.6 | 8B |
| Llama-8B-Instruct-3.0 | 35.4 | 8B |
| Mistral-7B-Instruct-v0.2 | 30.4 | 7B |
| Llama-3B-Instruct-3.2 | 22.2 | 3B |
| Mistral-7B-Instruct-v0.3 | N/A | 7B |
| DeepSeek-R1-LLama-1.5B | N/A | 1.5B |
| DeepSeek-R1-LLama-14B | N/A | 14B |
| Qwen2-VL-72B-Instruct | N/A | 72B |

Table 8: A complete list of models tested for this experiment along with their MMLU-PRO scores as publicly published. Parameter counts are listed where publicly available.
Meta Platforms; Meta AI (2024a;b); Team (2024a); Jiang et al. (2023); Mistral AI (2024); Team (2024b); Anthropic (2024); OpenAI (2024); Wang et al. (2024)

8 show's the Cramer's V for each contextual category and model, with Topic consistently having the largest effect on decisions for all models except for GTP4o, followed by actor type and finally world type.

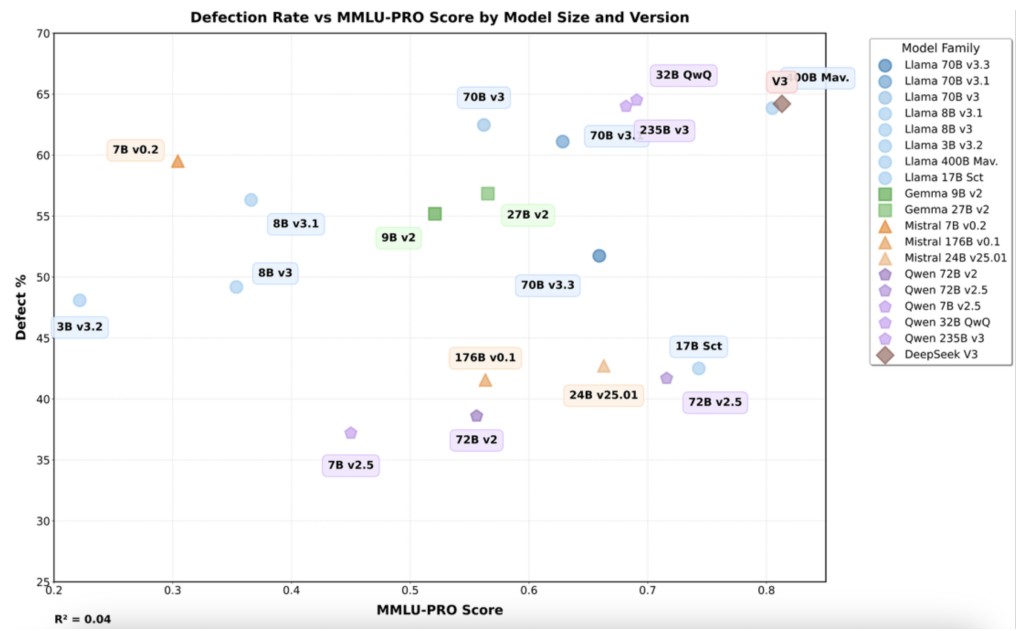

Figure 7: Defection rate plotted against MMLU score. Note that each shape represents a different model family.

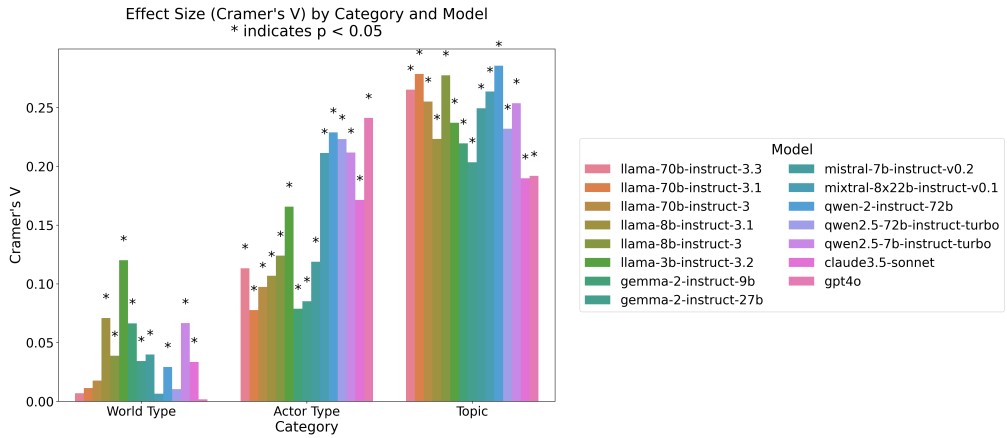

Figure 8: Effect size per model and per topic as measured by Cramer's V. Across model families and sizes, Topic consistently has the largest effect on decision, with the only outlier being GPT4o for which Actor Type has the largest impact.

## F GAME RECOGNITION

In response to each scenario, Claude, Llama, and GPT-4 were required to provide both a decision and its justification. The models frequently incorporated game theory concepts in their reasoning, though the frequency varied considerably by model (ranging from 16.5% to 70.5%). These percentages were calculated by using Claude 3.5 Sonnet to analyze the reasoning chains. For each justification, Claude was prompted with the following question:

PROMPT: DOES THIS TEXT EXPLICITLY MENTION THE PRISONER'S DILEMMA OR GAME THEORY? RESPOND ONLY WITH <YES> OR <NO> FOLLOWED BY THE RELEVANT SENTENCE(S). HERE IS THE TEXT: TEXT

The models' tendency to reference game theory suggest they recognize these scenarios as formal games and use this framework in their decision-making process. In Appendix F, we show how

the cooperate/defect proportion changes depending on whether game theory was mentioned in the justification. We see small, yet non-trivial changes to the proportion of cooperation.

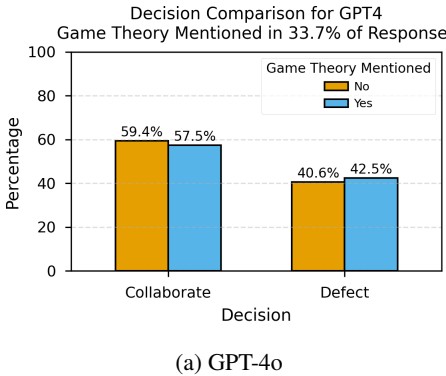

(a) GPT-4o

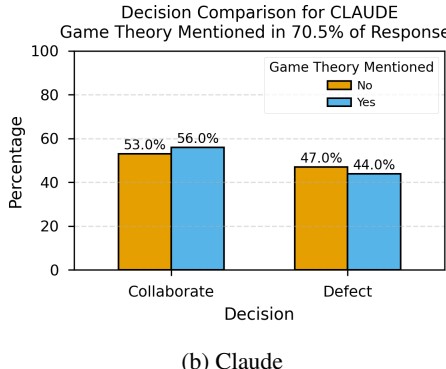

(b) Claude

## G  HEATMAPS

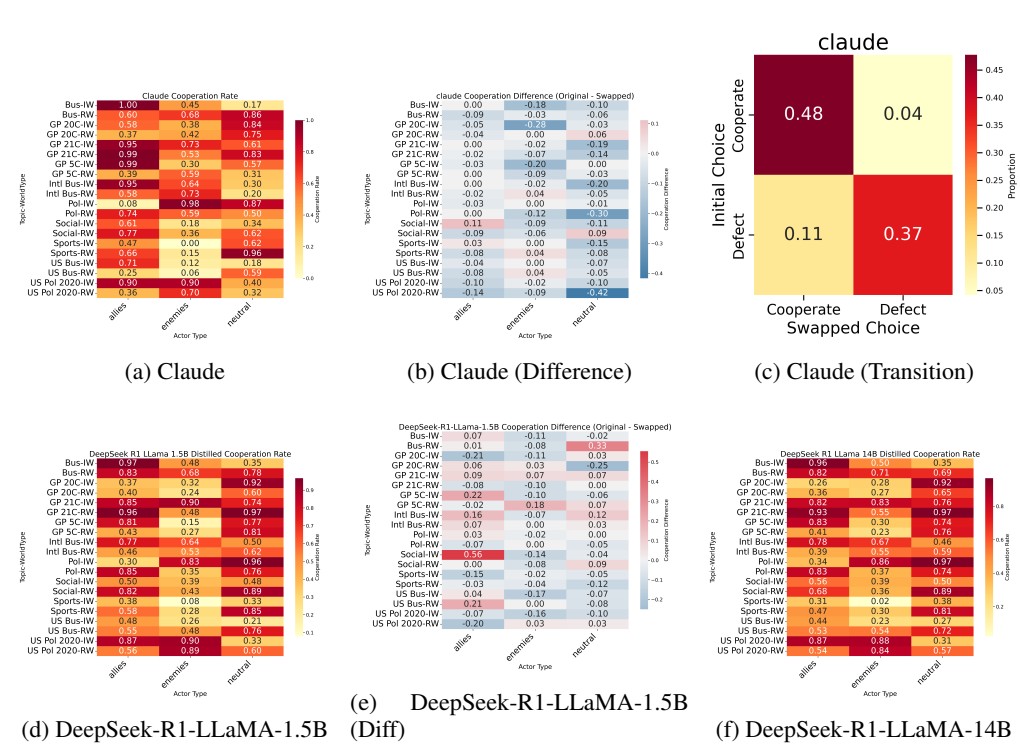

Figure 10: Distribution of decisions: Claude and DeepSeek-R1 models.

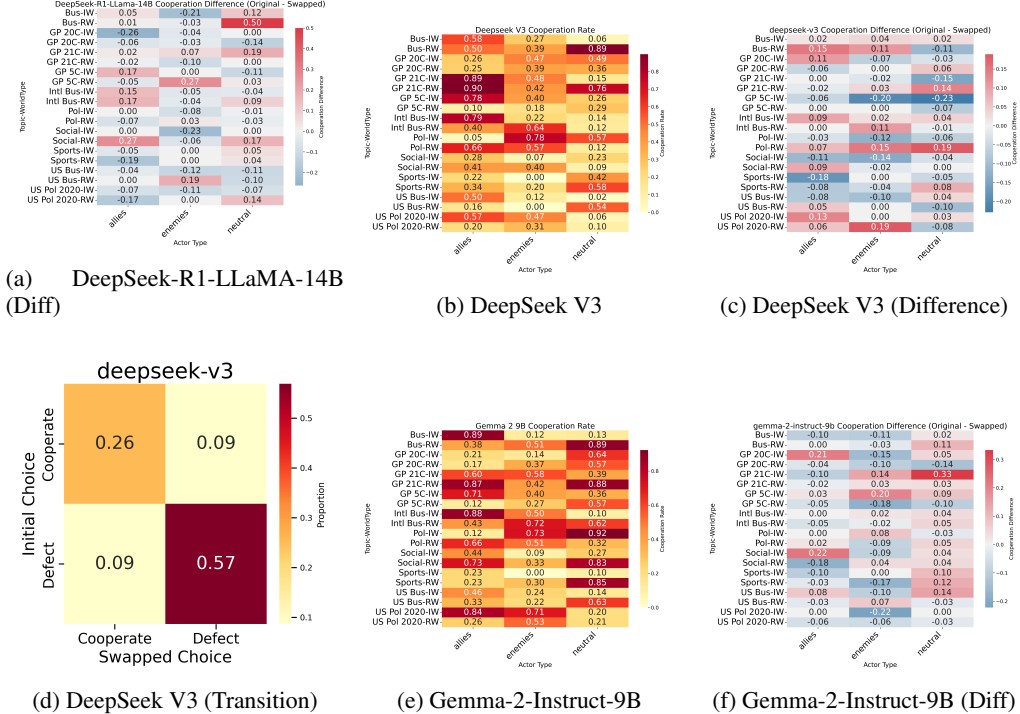

Figure 11: Distribution of decisions: DeepSeek and Gemma-2-9B models.

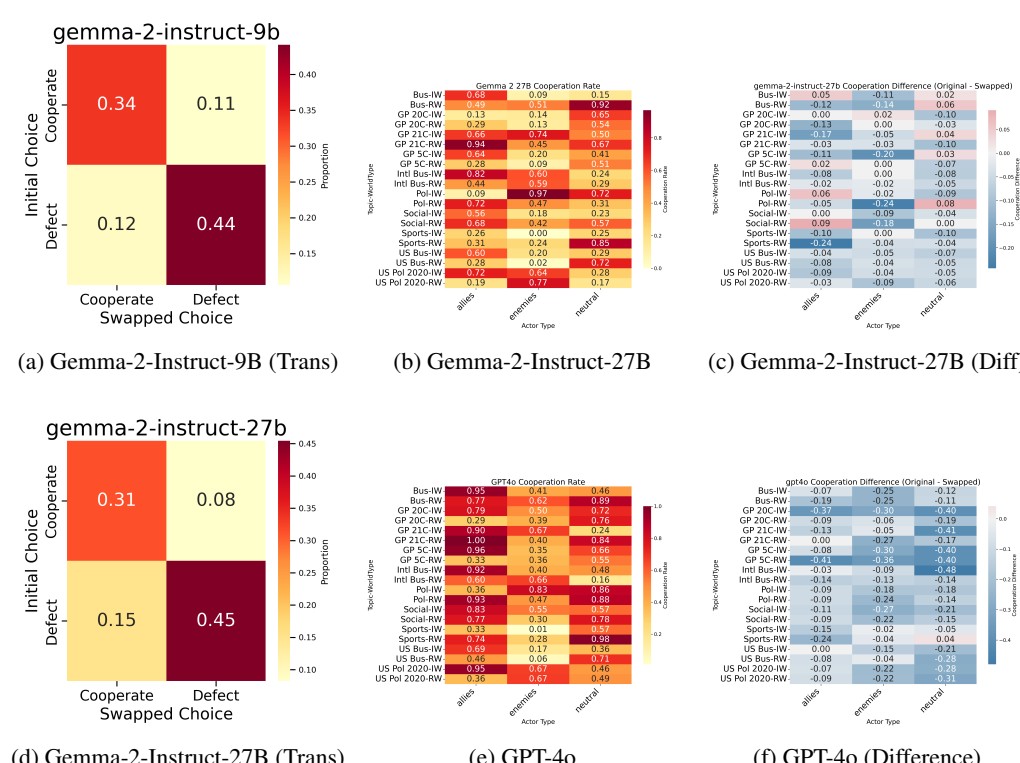

Figure 12: Distribution of decisions: Gemma-2 and GPT-4o models.

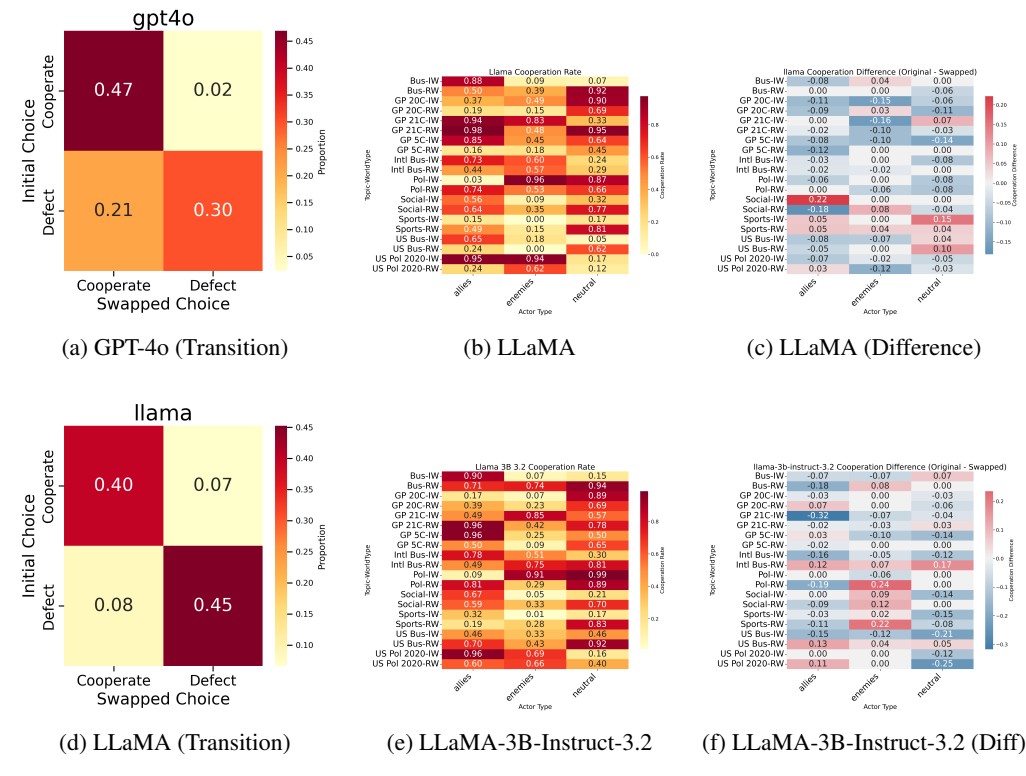

Figure 13: Distribution of decisions: GPT-4o and LLaMA base models.

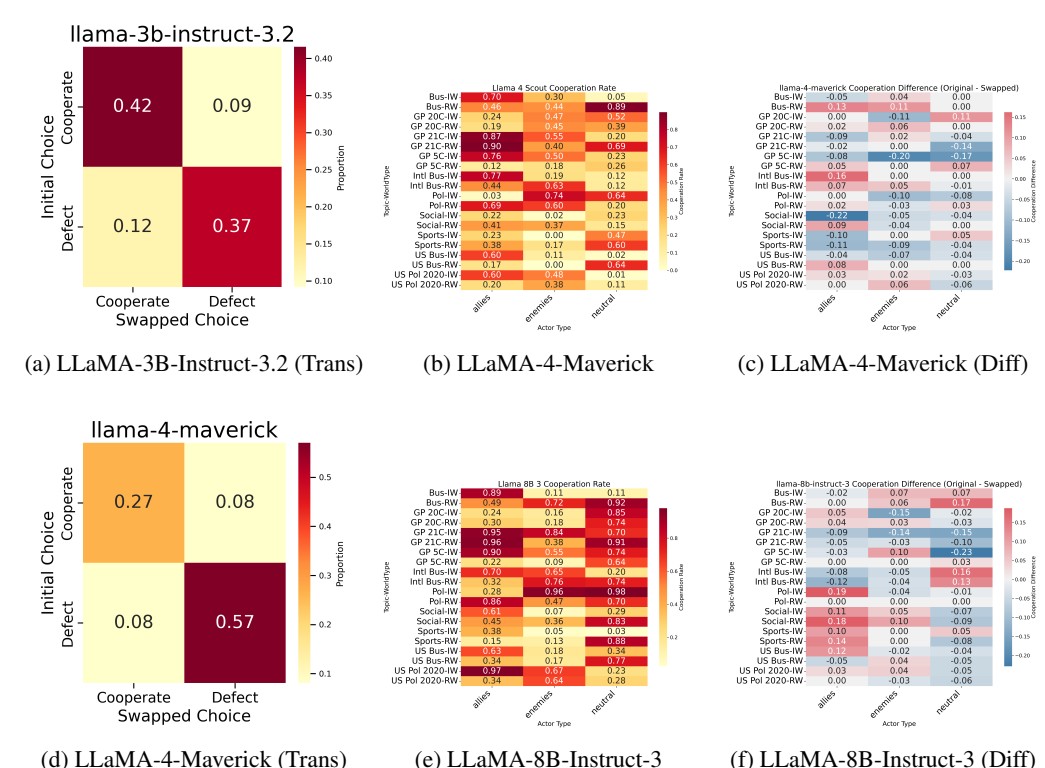

Figure 14: Distribution of decisions: LLaMA-3B, LLaMA-4-Maverick, and LLaMA-8B-3 models.

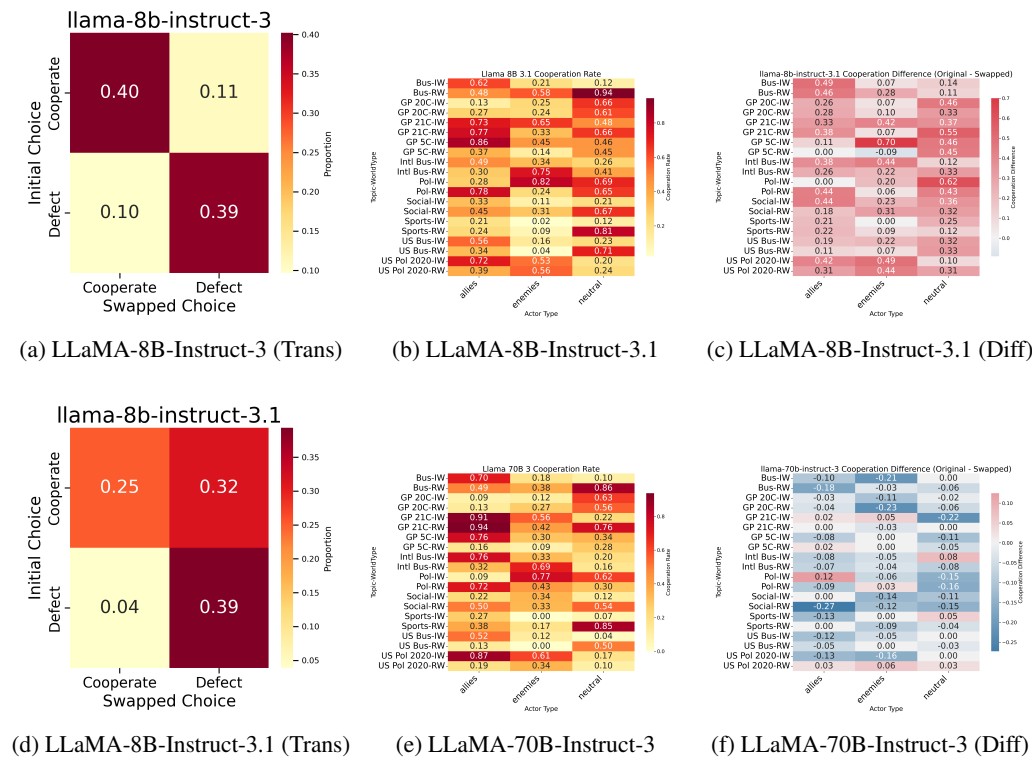

Figure 15: Distribution of decisions: LLaMA-8B-3.1 and LLaMA-70B-3 models.

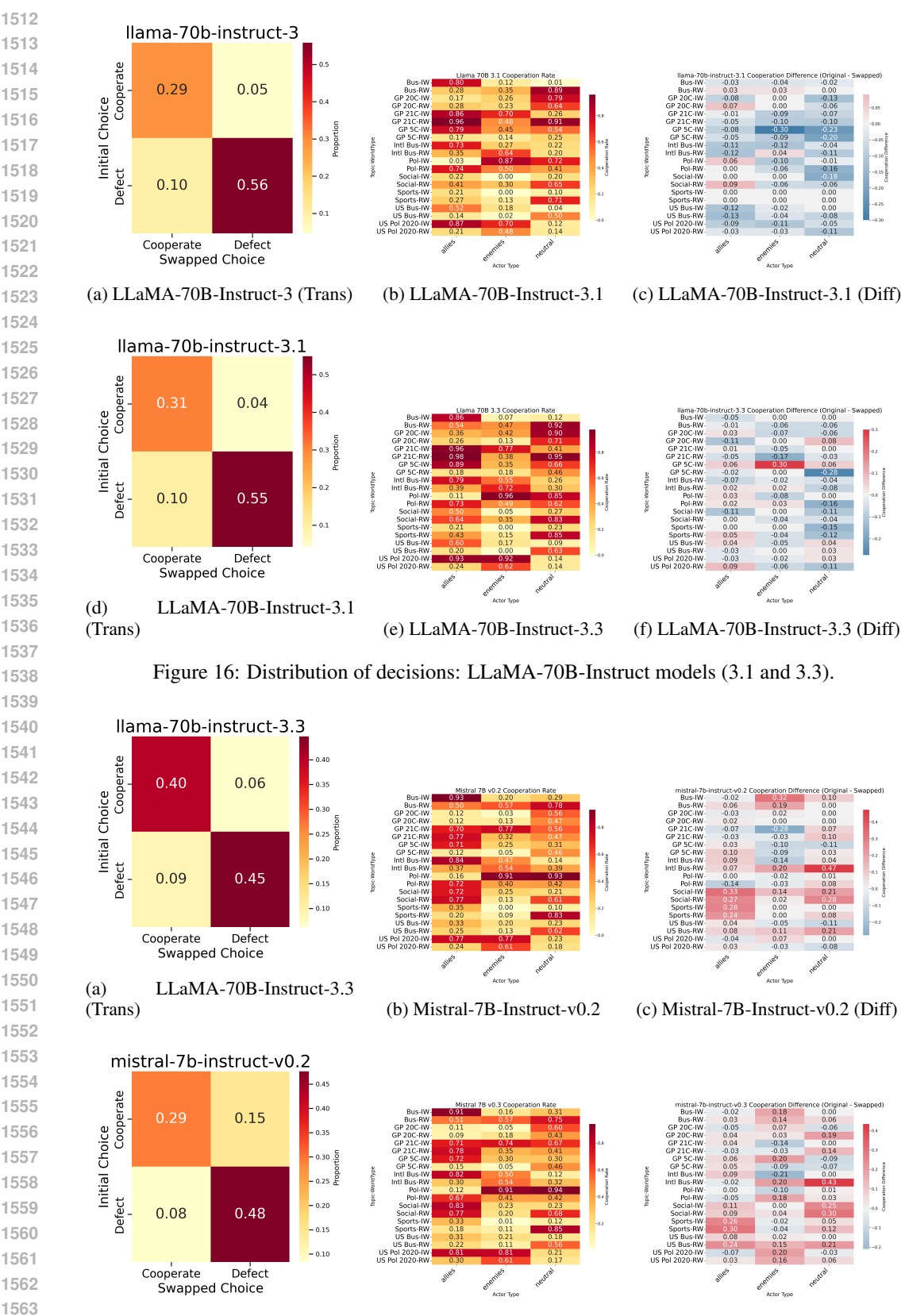

(a) LLaMA-70B-Instruct-3 (Trans)   (b) LLaMA-70B-Instruct-3.1   (c) LLaMA-70B-Instruct-3.1 (Diff)

(d)   LLaMA-70B-Instruct-3.1
(Trans)   (e) LLaMA-70B-Instruct-3.3   (f) LLaMA-70B-Instruct-3.3 (Diff)

Figure 16: Distribution of decisions: LLaMA-70B-Instruct models (3.1 and 3.3).

(a)   LLaMA-70B-Instruct-3.3
(Trans)   (b) Mistral-7B-Instruct-v0.2   (c) Mistral-7B-Instruct-v0.2 (Diff)

(d) Mistral-7B-Instruct-v0.2 (Trans)   (e) Mistral-7B-Instruct-v0.3   (f) Mistral-7B-Instruct-v0.3 (Diff)

Figure 17: Distribution of decisions: LLaMA-70B-3.3 and Mistral-7B models.

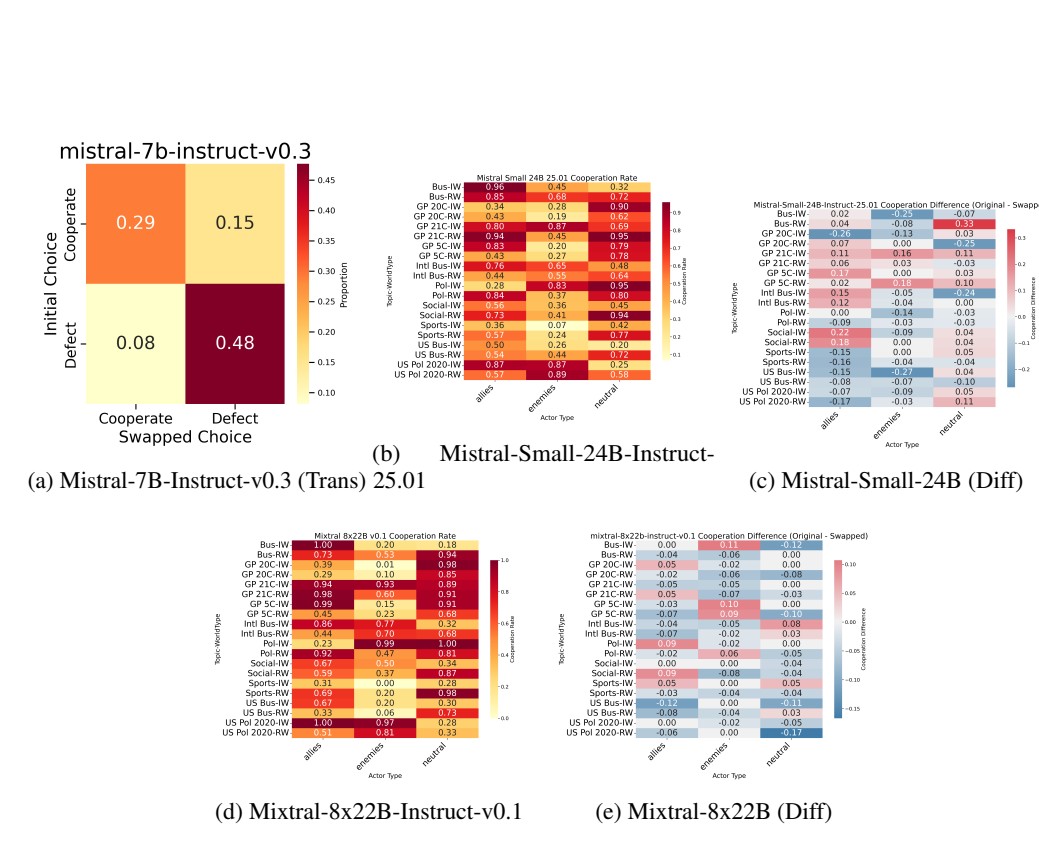

(a) Mistral-7B-Instruct-v0.3 (Trans)

(b) Mistral-Small-24B-Instruct-25.01

(c) Mistral-Small-24B (Diff)

(d) Mixtral-8x22B-Instruct-v0.1

(e) Mixtral-8x22B (Diff)

Figure 18: Distribution of decisions: Mistral-Small-24B and Mixtral-8x22B models.

