# OpenReview forum: "Framing the Game: How Context Shapes LLM Decision-Making"
_ICLR.cc/2026/Conference — Submitted to ICLR 2026_

### Official Review · Reviewer_mVh3 · 2025-10-20

**Soundness:** 2
**Presentation:** 2
**Contribution:** 3
**Rating:** 4
**Confidence:** 4

**Summary:**

This paper studies how narrative framing and context affect LLM decision-making in Prisoner's Dilemma scenarios. The authors develop an LLM-based generation framework to create varied vignettes with a consistent underlying Prisoner's Dilemma structure under real-world and imagined modes, varying topics and actor relationships. They examine 6 LLMs and find a significant context-dependent decision-making phenomenon that is largely predictable from these variables.

**Strengths:**

1. **Novel and meaningful perspective.**
This paper discusses an underexplored but fundamental question: how contextual framing shapes LLMs' decision-making, which provides a refreshing view for analyzing LLM rationality.

2. **Well-structured and scalable framework.**
The proposed vignette generation pipeline is scalable, flexible, and reproducible. By systematically varying contextual dimensions under the payoff matrix, the authors create a modular evaluation framework that can be easily extended to new games or social settings, allowing controlled and large-scale evaluation.

**Weaknesses:**

1. **Lack of deeper interpretation of contextual influence in decision-making.**
The experimental results show that actor type strongly influences cooperation rates (Allies > Neutral > Enemies, according to Figure 3), which the authors interpret as evidence of contextual instability in LLM decision-making. However, such variance could also be explained as an expression of human-like social preferences (e.g., trust and reciprocity) that naturally affect the cooperative behavior in real-world interactions, even though humans often deviate from the Nash equilibrium when facing allies or friends, reflecting contextual priors rather than irrationality.  Therefore, the observed pattern might not indicate poor rational consistency but reveal a contextually guided reasoning that LLMs exhibit implicit priors about social trust (e.g., allies → more cooperation; enemies → more defection). I think the authors can explore this interpretation and discuss whether an ideal LLM should adhere to the Nash equilibrium or emulate human social reasoning. If the authors can include human experiments and evaluate the alignment of LLMs and humans would substantially deepen the theoretical impact of the paper.


2. **Limited robustness analysis.**
In GAMA-Bench [1], the authors conducted a generalizability experiment by varying the payoff parameters and found that some models' decision-making patterns changed under different conditions. In this paper, the authors preserve a consistent payoff matrix (Figure 1) to generate those vignettes, but the contexts rarely provide the explicit and quantitative payoff descriptions. In Appendix A, only 1/8 vignettes specify the numerical and comparable gains, while others use affective terms like "low/high level of happiness". This raises two concerns for me: (1) Without explicit and comparable payoffs, the cooperation rates might be varied across different contexts, especially given that LLMs' decision-making does not exactly follow the Nash Equilibrium. (2) I still have a question about whether the observed behavioral variance reflects contextual sensitivity or noise introduced by inconsistent payoff.

3. **Lack of behavioral analysis.**
Sections 3.3.1 and 3.3.2 focus heavily on the cooperation proportions across topics, world types, and actor types, but the analysis is mostly descriptive. For example, the authors repeatedly restate surface-level variance patterns (e.g., cooperation in GP 21C > GP 5C, allies > neutral > enemies) without deeper behavioral interpretation. Their explanations remain high-level speculations (e.g., contextual influence and training-data overlap). I think a stronger contribution is to connect these phenomena to evidence from the models' reasoning traces, identifying which cues drive cooperation or defection biases with ablation experiments to evaluate such hypotheses. This helps the study turn from a report into a behavioral-mechanistic analysis, improving both rigor and insight.
The results in Section 3.3.3 provide a good motivation for studying the behavioral-mechanistic, because the authors show that models' decision-making is not random. However, this section does not provide insight into why these variables matter or what linguistic and moral features affect the predictions.

[1] How Far Are We on the Decision-Making of LLMs? Evaluating LLMs' Gaming Ability in Multi-Agent Environments. Huang et al.

**Questions:**

1. Cross-gaming Generalization: How does this phenomenon perform on other types of classical economy games (e.g., Public Goods Game), and do LLMs exhibit similar patterns?
2. Robustness Evaluation: How does the cooperation rate vary under different prompts (e.g., using the model to rephrase the generated context) and different payoff matrices?
3. Mechanistic and Behavioral Insights: What are the critical reasons or cues (e.g., moral, responsibility, or pragmatic) that lead to cooperation or variance? Any ablation-based analysis to support your claim?
4. Figure 2b) Should the arrows "Send decision prompt" and "Return decision (A/B)" reach "Decision LLM"? If not, what is "Decision LLM"?

---

> ### Author Response · Authors · 2025-11-21
>
> We thank the reviewer for their precise and thoughtful critiques.
>
> Human-like Preferences
>
> On the point on "human-like social preferences", we think this is an excellent possible interpretation of the results. We think part of the value of this work is that we demonstrate these preferences are captured whether we want them to be or not (for example, the models being more likely to collaborate when dealing with modern political scenarios rather than ancient ones). We think this framework is valuable as it easily and inexpensively elicits which of these human-like social preferences models have captured, which can spawn future analysis or adjustments of pre and post-training data and techniques. For example, we think that future work could explore whether there is some feature that separates representations of modern political figures from older ones and that explains the difference in cooperation rates between those two contexts.
>
> We present this work as valuable in that it can spawn many of these further analyses, and we think the introduction of this framework for eliciting these behaviors, along with showing that even without looking at the model weights, these behaviors are largely predictable not even from the prompt embedding, but from a set of just three contextual tags is of interest.
> We agree, though, that technically speaking, the identification of specific features or steering vectors would be fascinating, and hope to include a mechanistic-focused analysis in followup work.
>
> Ordinal vs Numeric Payoffs
>
> On the lack of explicit payoffs, we deliberately chose ordinal/descriptive payoffs (e.g., "moderate success" vs. "significant failure") to simulate real-world ambiguity. When we tested explicit numbers (e.g., "Gain $500"), models would more often notice the game structure and would explicitly calculate the Nash Equilibrium. We were more interested in model behavior in scenarios where there is no explicit payoff, which we think captures a wide variety of decision-making situations under uncertainty in politics and business.

---

> > ### Comment · Reviewer_mVh3 · 2025-11-25
> > **Response to authors rebuttal**
> >
> > This rebuttal does not address my concerns.
> >
> > 1. The authors interpret the observed behaviors as "human-like social preferences" without any human alignment study to support it. I feel this interpretation is somewhat overstated.
> >
> > 2. In weakness 1, I emphasized that only reporting the cooperation rate variance is not enough. The authors should investigate the reasons behind these patterns (e.g., why more collaboration under political scenarios). I expected the authors to provide a deeper, reasoning-based analysis (e.g., linguistic cues or moral reasoning signals). Still, they simply restated that "these preferences are captured" without any additional insight.
> >
> > 3. I understand the motivation for descriptive payoffs. However, my concern was whether the description of the payoff affects the reliability of conclusions. I expected ablation experiments comparing explicit numeric payoffs vs descriptive payoffs, or at least prompt-robustness evaluation.
> >
> > Overall, the rebuttal reiterates the paper's original claims without providing any further clarification or answering the questions I raised. My concerns remain unresolved.

---

### Official Review · Reviewer_mQG5 · 2025-10-24

**Soundness:** 3
**Presentation:** 3
**Contribution:** 2
**Rating:** 6
**Confidence:** 3

**Summary:**

Large language models are increasingly used for decision support, yet evaluations often overlook how context framing affects apparent rationality. This study presents an evaluation framework that systematically varies key features and procedurally generates vignettes to create diverse scenarios. Comparing decisions across contexts with identical game structures reveals strong context effects on model choices—variability that is largely predictable but highly sensitive to framing. The results highlight the need for dynamic, context-aware evaluation to ensure reliable real-world deployment and offer initial guidance for its design.

**Strengths:**

- The paper is well written and logically clear. Even readers who are not familiar with this field can understand it.

- The experiments are comprehensive and well-designed. They include multiple models, topics, and metrics, and cover both model behavior and the reasons behind behavior changes.

- For me personally, “How context shapes LLM decision-making” is a very interesting and practical topic. If we want to deploy LLMs widely in real-world production, they will face much more complex contexts than today. Understanding how the model’s decisions change is very important.

**Weaknesses:**

1. The paper lacks enough technical contribution. The authors demonstrate that context framing has a significant impact on LLM responses in the Prisoner’s Dilemma and advocate for dynamic evaluation strategies. Their evaluation framework (Figure 2) appears to use existing LLMs to generate different contexts.

2. As the authors note in the limitations, the decision problem studied is simplified. This helps build an analysis framework, but it makes the real-world applicability of the conclusions unclear. Adding more realistic decision experiments (e.g., an agent in a company or bank setting) would help address this.

3. Reasoning models are now an important part of LLMs, but the paper’s evaluation of reasoning models seems very limited and lacks discussion. This makes the work feel incomplete at the current time.


Typo： Line 38 “of the art LLMs. including” → “of the art LLMs, including”

**Questions:**

See Weakness.

---

> ### Author Response · Authors · 2025-11-21
>
> We thank the reviewer for their precise and thoughtful critiques, and will fix the noted typo.
>
> Technical Contribution
>
> On the lack of technical contribution, as noted above, our contribution is identifying and quantifying a failure mode (context sensitivity) and providing a reproducible, inexpensive pipeline to test it. The finding that this sensitivity is systematic and predictable based not just on the prompt itself or the model weights, but on a set of simple tags identifying the prompt’s context we think is an admittedly incremental but novel insight into model behavior. Moreover, the fact that one could use this methodology in the discovery phase for model biases with little to no human intervention and cost we think is helpful for spurring community research into exactly why models have these sensitivits and how to fix them should we decide it is necessary to do so.
>
> Applicability to Agentic Settings
>
> As for the simplified setting, our vignettes (e.g., "Global Politics," "Business Competitors") simulate the decision node agents would face in the real world, the difference being this is a single shot API call without an agent loop. We believe our results are not incopaitble with an agentic workflows – the context framing is meant to simulate the context an agent would have from the message history or memory artifacts.
>
> Reasoning Models
> We agree reasoning models are an important part of modern LLMs, and tested 2 small distillations of the DeepSeek R1 reasoning models, see Figures 10 and Figure 6B ablation results. We saw similar trends in defection and predictability.

---

> > ### Comment · Reviewer_mQG5 · 2025-11-24
> >
> > Thanks for your reply.

---

### Official Review · Reviewer_bXkZ · 2025-10-27

**Soundness:** 2
**Presentation:** 2
**Contribution:** 1
**Rating:** 2
**Confidence:** 4

**Summary:**

This paper proposes a framework for evaluating how framing affects LLM behavior in strategic decision-making. Specifically, it instantiates a two-player, one-shot Prisoner’s Dilemma, uses Factorial-Survey–inspired procedural vignette generation, and assesses predictability with lightweight feature and embedding models (e.g., XGBoost).

**Strengths:**

1. **Clear formalization** with a strictly dominant-strategy Prisoner's Dilemma gives a neat normative baseline.
2. **Factorial Survey to procedural generation pipeline** is articulated and reasonably motivated for stress-testing contextual sensitivity.
3. The paper **reports predictable framing effects and family-level differences**; e.g., correlation between MMLU and defection in some model families.

**Weaknesses:**

1. **Related work is narrow.** The literature review centers mostly on prior two-player static setups and does not adequately engage with the growing body of multi-player, multi-round/interactive evaluations. This weakens the motivation and overstates novelty relative to current trends.
2. **Limited generality and claims outstrip evidence.** All experiments instantiate a two-player, one-shot, symmetric 2×2 Prisoner’s Dilemma in normal form, with action set {Cooperate, Defect}; models are explicitly told not to consider repeated play. Despite describing the framework as supporting “an arbitrary, user-specified range of scenarios,” no evidence is shown beyond this single game class.
3. **Essentially a robustness/sensitivity study with limited methodological novelty.** The central result, where framing drives behavior; behavior is somewhat predictable—is interesting but familiar. Technically, predictability uses off-the-shelf XGBoost and standard sentence embeddings, with routine grid search. This reads more like a baseline robustness analysis than a new evaluation methodology.
4. **LLM-based QC is reasonable in principle, but the current pipeline relies on a single judge (GPT-4o) with no human agreement, judge-swap, or sensitivity analyses.** As a result, it is unclear whether findings are robust to reasonable alternatives or partly circular (e.g., upstream biases encoded by the judge).

**Questions:**

1. If extended to multi-player/multi-round games or to non-dominance structures (e.g., Stag Hunt, Chicken, public-goods, voting), which findings do you expect to persist, and which do you expect to flip?
2. When a model recognizes the Prisoner's Dilemma yet behaves differently across framings, is that due to values, instruction-following vs. rationality trade-offs, or sampling noise? Any discriminative analysis planned?
3. Please detail your QC. As described, QC is LLM-based only, provide judge-swap, threshold sensitivity and generator–judge decoupling, and report coverage metrics and leakage checks for game-structure cues.

---

> ### Author Response · Authors · 2025-11-21
>
> We thank the reviewer for their precise and thoughtful critiques.
>
> Related Work and Contextual Framing
>
> On the narrowness of the related work, our work focuses specifically on holding the game constant and varying the contextual framing, rather than changing the underlying game like https://arxiv.org/abs/2411.05990. While this contextual framing could be applied to multi-round games or games with more players, we focused on the simplest game structure as the results for even this trivial game showed a large and predictable impact of context on strategy, even when the dominant strategy is abundantly clear.
> We think that being able to procedurally elicit contextual biases in the model using a grounding in game theory is a real contribution.
>
> Predictability and Methodology
>
> We also demonstrate that we can predict an LLM’s complex decision using a lightweight classifier on not only the prompt embedding but the small set of generative contextual tags. This implies that these biases are predictable with orders of magnitude less compute than the model itself, and with less information than a linear probe, or a prompt classifier would have access to, which we have not seen discussed before in the litterature. Moreover, we think the automatic analysis of contextual biases using groudning in the same game structure as the controlled variable is a simple insight but still a novel contribution. We recognize though that the resulting biases are quite intuitive, and that the methods used were simple and thus may be insufficient for a conference of this caliber.
>
> Quality Control and Use of LLM Judges
>
> Similar to the above, we acknowledge the concern of using an LLM judge to validate our vignettes. To mitigate this, we employed a rigorous rubric (Appendix B) as is accepted best practice with spot-checked results. While a widely used methodology, we agree this is a fundamental limitation, though we do not think it detracts from the usefulness of the generative framework.

---

> > ### Comment · Reviewer_bXkZ · 2025-11-28
> >
> > Thank you for the reply. While I appreciate the clarifications, they do not sufficiently resolve my main concerns. I had expected the response to provide stronger evidence or analysis such as:
> >
> > (1) clearer engagement with the broader game-theoretic evaluation literature
> >
> > (2) some demonstration of applicability beyond the single PD setup
> >
> > (3) robustness checks addressing the reliance on a single LLM judge
> >
> > These points are central to the paper’s claimed generality, novelty, and methodological soundness. As these issues remain largely unaddressed, I will maintain my original score.

---

### Official Review · Reviewer_aAtL · 2025-11-02

**Soundness:** 2
**Presentation:** 2
**Contribution:** 2
**Rating:** 4
**Confidence:** 4

**Summary:**

This paper introduces a novel, dynamically generated evaluation framework for probing how narrative context influences large language models (LLMs) in one-shot Prisoner’s Dilemma (PD) games. By procedurally generating thousands of varied vignettes along three axes—Topic (e.g. US politics, business, historical eras), World Type (real vs. imaginary), and Actor Relationship (allies, enemies, neutral)—the authors isolate framing effects while holding the underlying 2×2 PD payoff matrix constant. They evaluate 25 LLMs (including GPT-4o, Claude 3.5, Llama-3.3, etc.), demonstrate significant context-dependent variance in cooperation rates, show that these variances are largely predictable via simple XGBoost classifiers (using either contextual metadata or vignette embeddings), and analyze inter-model agreement, positional biases, and trends with benchmark capability (MMLU-Pro). They argue for more robust, context-aware evaluation protocols and release their vignette-generation code for reproducibility.

**Strengths:**

1. This paper conducts experiments on a wide range of open- and closed-source LLMs across diverse contexts, providing strong evidence that framing systematically alters model behavior.

2. Interesting topic and findings.

**Weaknesses:**

1. The paper focuses solely on the one-shot PD. While foundational, it remains unclear how generalizable the methodology and findings are to other strategic or decision-theoretic settings (e.g. coordination games, sequential dilemmas, multi-player interactions).

2. Vignette quality and neutrality are assessed automatically via GPT-4o using a 3-dimensional rubric. This introduces a potential circularity: an LLM judging its own procedural outputs. Reporting a small human validation sample to calibrate and validate rubric thresholds can address this issue.

3. While the paper documents that framing effects arise, it does not deeply probe why certain contexts drive higher cooperation or defection

**Questions:**

1. Have you experimented (or do you plan to) extending your framework beyond the one-shot Prisoner’s Dilemma to other canonical games (e.g. Stag Hunt, Hawk-Dove, public goods)?

2. The rubric for PD structure, clarity, and bias-neutrality is applied automatically by GPT-4o. Have you conducted any human evaluations to confirm the rubric’s judgments？

3. Your XGBoost models show that topic and actor type are strong predictors of cooperation. Can you illuminate why certain framings sway the models?

---

> ### Author Response · Authors · 2025-11-21
>
> We thank the reviewer for their thoughtful critiques.
>
> Game Selection and Scope
>
> We focused on the one-shot PD as in the family of single-shot two-player games, it clearly isolates the tension between individual rationality (dominant strategy: Defect) and collective welfare (Cooperate). Other games listed in standard taxonomies lack this specific tension.
>
> For example, since stag hunt has both a risk-dominant and payoff-dominant strategy, it measures more the risk-attitude of these models, rather than their propensity to collaborate. Hawk-Dove is interesting, and had we seen no variation in the PD games, we were planning on adding something like Hawk-Dove where to be rational the model would actually need to calculate explicit payoffs, rather than relying on just strict ordering of outcomes. However, even using this most basic game structure (PD), we saw large amounts of variance, supporting our claim on the contextual dependence of collaborative behavior.
>
> Quality Control and Use of LLM Judges
>
> We acknowledge the concern of using an LLM judge to validate our vignettes. To mitigate this, we employed a rigorous rubric (Appendix B) rather than open-ended evaluation, with spot-checked results. While a widely used methodology, we agree this is a fundamental limitation, though we do not think it detracts from the usefulness of the generative framework.
>
> Scope and Mechanistic Analysis
>
> Our primary contribution is the evaluation methodology and the empirical evidence that behavioral variance is predictable based on context features and embeddings, even without mechanistic methods. While something like mechanistic work we agree would be fascinating, we would be limited in our explanations to the scope of a small subset of open source models, and any findings of specific features controlling these outputs would likely be model specific.
>
> We actually think the main causal reason we see this predictability is because of the shared training data across model families. We see the value of this work as a way to explore and elicit these behaviors (like the models having a more less aggressive stance in modern politics than ancient politics) with automatic, relatively cheap and fast inference patterns. Once elicited, researchers can examine their model families and biases of choice, exploring mechanistically why a specific model is more collaborative in X type of scenario than Y type scenario.

---

### Meta-Review · Area_Chair_9hK9 · 2026-01-06

**Summary:**

This paper proposed a novel evaluation framework to study the impact of context framing in strategic decision-making under Prisoner's Dilemma scenarios. The framework allows to generate different evaluation scenarios by varying different topics, world settings, and actor relationships. Different LLMs are evaluated, showing high-levels of context dependency, which can be predicted with simple method XGBoost.

The paper presents an interesting study to evaluate the influence of context for decision making. However, there are a few important limitations raised by the reviewers:

(1) The paper focuses solely on the one-shot PD. It is difficult to see the generalizability of the proposed framework to other decision making scenarios.

(2) Vignette quality and neutrality are simply evaluated by a single LLM without human evaluation.

Due to these limitations, the paper is not ready for publication at ICLR.

**Reviewer Concerns:**

Reviewer aAtL and bXkZ both raised the potential generalization capability of the proposed framework in different decision making scenarios, which were not addressed by the rebuttal.

The evaluation problem by a single LLM, raised by reviewer aAtl and bXkZ are also not addressed.

**Reviewer Scores:**

Due to the above two limitations, it is difficult to convince the reviewers to change their scores.

---

### Decision · Program_Chairs · 2026-01-26

Reject